



# Future changes in North Atlantic winter cyclones in CESM-LENS. Part I: cyclone intensity, PV anomalies and horizontal wind speed

Edgar Dolores-Tesillos[1], Franziska Teubler[2], and Stephan Pfahl[1]

[1]Institute of Meteorology, Freie Universität Berlin, Berlin, Germany
[2]Johannes Gutenberg-Universität Mainz, Mainz, Germany

**Correspondence:** Edgar Dolores-Tesillos (edgar.dolores@met.fu-berlin.de)

**Abstract.** Strong low-level winds associated with extratropical cyclones can cause substantial impacts on society. The wind intensity and the spatial distribution of wind maxima may change in a warming climate; however, the involved changes in cyclone structure and dynamics are unclear. Here, such structural changes of strong North Atlantic cyclones in a warmer climate close to the end of the current century are investigated with storm-relative composites based on Community Earth System Model Large Ensemble (CESM-LENS) simulations. Furthermore, a piecewise potential vorticity inversion is applied to associate such changes in low-level winds to changes in potential vorticity (PV) anomalies at different levels. Projected changes in cyclone intensity are generally rather small. However, using cyclone-relative composites, we identify an extended wind footprint southeast of the center of strong cyclones, where the wind speed tends to intensify in a warmer climate. Both an amplified low-level PV anomaly driven by enhanced diabatic heating and a dipole change in upper-level PV anomalies contribute to this wind intensification. On the contrary, wind changes associated with lower- and upper-level PV anomalies mostly compensate each other upstream of the cyclone center. Wind changes at upper levels are dominated by changes in upper-level PV anomalies and the background flow. All together, our results indicate that a complex interaction of enhanced diabatic heating and altered non-linear upper-tropospheric wave dynamics shape future changes in near-surface winds in North Atlantic cyclones.

## 1 Introduction

Extratropical or mid-latiude cyclones strongly modulate weather and climate in the North Atlantic region. For example, as the cyclones control cloud amount, they regulate the radiation received by the extratropics. Storm tracks are the areas where extratropical cyclones are most likely to form, propagate, dissipate, as well as contribute the most energy and momentum transport (Chang et al., 2002; Shaw et al., 2016). Extratropical cyclones passing across the midlatitudes generate specific weather conditions (Catto, 2016). Cyclones and associated fronts produce locally up to 90% of annually accumulated precipitation (Catto et al., 2012; Hawcroft et al., 2012) and also contribute significantly to extreme precipitation events (Pfahl et al., 2014; Catto and Pfahl, 2013). In addition, cyclones can lead to the formation of strong winds (Browning, 2004; Leckebusch et al., 2006). The economic losses due to these winds have been shown to scale approximately with the third power of the near-surface wind speed (Roberts et al., 2014). The strongest near-surface winds typically occur in the dry intrusion region behind the cold front





and the cold conveyor belt, sometimes associated with sting jets (Clark and Gray, 2018), as well as in the warm region to the south-east of the storm center, ahead of the cold front (Hewson and Neu, 2015; Shaw et al., 2016; Slater et al., 2017).

Several conceptual models have been developed to describe cyclone types and structures, such as the Norwegian cyclone model (Bjerknes, 1919) and the Shapiro and Keyser (1990) conceptual model. Airstreams linked to cyclones are described in a three-dimensional conceptual model that incorporates warm and cold conveyor belts (Browning, 1990; Wernli and Davies,
1997; Madonna et al., 2014). A helpful variable to analyze mid-latitude atmospheric dynamics and the development of cyclones is Potential Vorticity (PV) (e.g. Hoskins et al., 1985). PV is a conserved quantity under adiabatic conditions; its rate of change is then determined by advection. Any further changes can be attributed to the generation or destruction of PV by non conservative processes, like diabatic processes (latent heating (LH) and radiative heating) and frictional forcing (Catto, 2016; Bluestein, 1992). A qualitative framework to understand the relation between PV and extratropical cyclone dynamics and the role of LH
is through three anomalous components in the PV-associated cyclone circulation: a positive upper-tropospheric PV anomaly, a positive potential temperature anomaly at the surface, and a positive lower-tropospheric PV anomaly (Davis and Emanuel, 1991; Davis, 1992). The positive lower-tropospheric PV anomaly is primarily generated by diabatic processes, in particular, latent heat release during cloud formation (Ahmadi-Givi et al., 2004; Stoelinga, 1996; Büeler and Pfahl, 2017), and can thus be regarded as a proxy for the influence of such diabatic processes on cyclone dynamics. The relative contributions of these
anomalies to the wind field and the cyclone intensity can be determined by applying PV inversion techniques (Davis and Emanuel, 1991; Davis, 1992; McTaggart-Cowan et al., 2003; Tochimoto and Niino, 2016; Flaounas et al., 2021).

Anthropogenic climate change is expected to lead to alterations also in the spatial distribution and properties of extratropical cyclones. Projections from climate models have been analyzed to identify such cyclone changes in a warming climate (Ulbrich et al., 2009; Zappa et al., 2013; Raible et al., 2018; Sinclair et al., 2020). While future changes in the North Pacific and, in
particular, the Southern Ocean, are, to first order, characterized by a poleward shift of storm tracks (O'Gorman, 2010), the pattern of projected changes in the North Atlantic region is more complex (Zappa et al., 2013). Such changes in the spatial distribution and intensity of cyclones are thought to result from a "tug of war" (Shaw et al., 2016) between different processes, such as changes in horizontal temperature gradients and baroclinicity, vertical stability, tropopause height and latent heat release (O'Gorman and Schneider, 2008; Pfahl et al., 2015). In particular, the increase in atmospheric moisture content in a warming
climate results in an increase also of latent heat release and, therefore, might intensify cyclones. Such a potential intensification may, however, be compensated by other factors such as an increase in static stability (which is also linked to the moisture increase, in particular in lower latitudes where the atmospheric stratification is closer to a moist adiabatic) (Catto et al., 2019). One way to identify the influence of altered LH on cyclone dynamics is via the PV framework. As the PV anomaly in the lower troposphere is primarily related to diabatic processes, increased LH is expected to result in an intensification of this anomaly.
Accordingly, an increase in lower-tropospheric PV in midlatitude cyclones has been detected in idealized model simulations (Pfahl et al., 2015; Büeler and Pfahl, 2019; Sinclair et al., 2020) and climate change studies with regional models (Marciano et al., 2015; Michaelis et al., 2017; Zhang and Colle, 2018). Tamarin and Kaspi (2017) and Tamarin-Brodsky and Kaspi (2017), using PV inversion, showed that such diabatically induced low-level PV increases contribute to an enhanced poleward motion





of cyclones as the climate warms. Nevertheless, it is not clear how (diabatic) PV changes link to structural changes in other
impact-relevant cyclone properties such as low-level wind velocity. This is one of the central questions of the present study.

Generally, there is a decrease in extratropical cyclone winds projected in the Northern Hemisphere as the climate warms,
which reduces the wind hazards associated with the cyclones (Catto et al., 2019). Nevertheless, there may still be specific
regions that will experience an increase in wind hazards, such as central and western Europe (Mölter et al., 2016). In addition,
uncertainty in such estimated wind changes arises from the fact that global climate models cannot explicitly represent high
wind-producing mesoscale circulation structures such as sting jets (Martínez-Alvarado et al., 2018). Information on structural
changes in extratropical cyclone winds and other properties can be obtained with the help of composite analyses (e.g. Bengtsson
et al., 2009; Catto et al., 2010; Dacre et al., 2012; Yettella and Kay, 2017; Pinto and Ludwig, 2020). While idealized modelling
experiments point to a broadening of the cyclones' strong wind footprint (Sinclair et al., 2020), composite changes from other
regional model studies are relatively noisy (Michaelis et al., 2017), which might be partially related to limited statistics. Here
this will be addressed with the help of extensive ensemble simulations providing multiple samples also of extreme North
Atlantic cyclones and their future changes.

This study adopts a composite perspective to study future changes in PV anomalies and associated wind changes in North
Atlantic cyclones. The paper is organized as follows. In Sect. 2, we introduce the CESM-LENS dataset. Section 3 presents the
cyclone tracking scheme, composite technique and PV inversion (PPVI) method. In Sect. 4, results are presented of simulated
future changes in cyclone tracks, and the changes in properties of intense cyclones are studied in more detail using composite
analysis and PPVI. Conclusions are provided in Sect. 5.

## 2   Data

We use 6-hourly output from 10 members of the CESM-LENS-ETH model ensemble, which were restarted from Community
Earth System Model Large Ensemble (CESM-LENS) simulations (Kay et al., 2015). The members differ by a small random
perturbation to their initial air temperature field on the order of $10^{-14}$ K. Each member has a horizontal resolution of $0.94^o$ x
$1.25^o$, latitude and longitude, respectively. The vertical coordinate is a hybrid sigma-pressure system with 30 levels. Two periods are analyzed: 1990-2000 for the present-day climate and 2091-2100 for future climate. Historical forcing (Lamarque et al.,
2010) was applied in the present-day period and Representative Concentration Pathway 8.5 (RCP8.5) forcing (Meinshausen
et al., 2011; Lamarque et al., 2011) in the future period. The simulations have been re-run for these periods (using restart files
from the original simulations) to obtain more comprehensive six-hourly output fields, such as vertical velocity on model levels.
Note that the re-runs are not bit-identical to the original CESM-LENS simulations.

To assess the ability of CESM-LENS to simulate observed cyclone track density and statistics, the historical simulations
(1990–2000) are compared with the ERA-Interim reanalysis (Dee et al., 2011) for the period 1979-2010.





## 3 Methods

### 3.1 Cyclone identification

Cyclones are identified with an updated version of the sea-level pressure (SLP) contour method developed by Wernli and Schwierz (2006), as described in Sprenger et al. (2017)). Cyclone centers are defined as local minima in the SLP field. Starting from every SLP minimum, closed isobars are identified at intervals of 0.5 hPa, and the outermost contour that does not exceed 7500 km in length is used as the outer boundary of the system. Cyclones that are close to each other (e.g., binary cyclones with two local SLP minima) are merged as long as the outermost closed contour's length does not exceed 7500 km, and only the deepest SLP minimum of each cyclone is kept and used for cyclone tracking. Successive SLP minima are connected to form a cyclone track if they occur within a specific search area determined by the previous cyclone trajectory. As in Pfahl et al. (2015), only cyclone tracks are taken into account in the following with a minimum lifetime of 24h and an SLP difference of at least 1 hPa between the minimum and the outermost contour at each time step. Following Neu et al. (2013) the track density is defined as the number of tracks passing a grid cell (with repeated entries of the same track being counted as one). Then, a regridding procedure is applied to project the trajectories onto a regular $1^o$ x $1^o$ grid.

In this study, we focus on analysing future cyclone changes over the North Atlantic region [longitude: -100° to 40° and latitude: 30° to 90°] for the extended winter season (October-March). We associate a cyclone track to the domain if it has at least one location inside the domain. Therefore, our analysis is based on more than 35,800 storms in the 11 extended winters of present-day climate and more than 30,800 storms in the 10 winters of future climate, pooling the data from all 10 ensemble members.

### 3.2 Composite analysis

Cyclone-centered composites are created for the 10% strongest storms occurring over the North Atlantic region (denoted as intense cyclones in the following), selected by their maximum relative vorticity at 850 hPa averaged over a radius of 250 km around the cyclone center (SLP minimum). Various fields are composited for the time of maximum intensity in accordance with the 850 hPa relative vorticity (t=0), as well as previous hours (t=-24, -18, -12, -6 h) and later hours (t=+6, +12, +18, +24 h). A radial grid with the pole centered on the cyclone center is generated, covering a spherical cap with a fixed radius of $10^o$. Meteorological fields are then extracted to this new radial coordinate system and averaged over the chosen extratropical cyclones. The coordinate transformation reduces the distortion due to change in zonal grid spacing with latitude, as described in previous studies (Bengtsson et al., 2009; Catto et al., 2010). In some previous studies, this spherical cap has been rotated in the direction of cyclone propagation. We have tested such a rotation and decided not to use it because it introduced some noise in the composites (e.g., frontal structures appeared to be less apparent in the rotated compared to the unrotated composites, see supplementary Fig. S1).

A complementary perspective on cyclone dynamics is obtained by constructing PV anomalies and PV anomaly profiles. To this end, a PV climatology is defined as follows: Firstly, the daily mean PV for each calendar day is calculated at each grid point and each vertical level of the model, separately for present-day (1990-2000) and future (2091-2100) climate. Then, a



30-day window is selected relative to each cyclone position, and composites (as described previously) are obtained, averaged over these 30 days, on different interpolated pressure levels (1000, 925, 850, 750, 700, 600, 500, 400, 300, 250, 200, 150 hPa) from the daily PV dataset. PV anomalies are then defined relative to these 15-day averages. Finally, PV anomalies are spatially averaged in a radius of 2.5° around to the cyclone center to construct the PV anomaly profiles.

### 3.3 Piecewise PV inversion

Piecewise potential vorticity inversion (PPVI) can be used to examine the contributions of PV anomalies ($q'$) to the wind field of an extratropical cyclone (Davis and Emanuel, 1991; Tochimoto and Niino, 2016). For a suitable set of balance and boundary conditions, the invertibility of PV allows the derivation of wind and geopotential height fields from the PV distribution (Hoskins et al., 1985; Seiler, 2019)

With PPVI, $q'$ in extratropical cyclones is commonly partitioned into three layers: the surface layer, the lower atmospheric layer, and the upper atmospheric layer (Seiler, 2019). The inversion may then be utilized to investigate the relative contributions of the PV anomalies in these layers to the flow field at specific levels (Tamarin-Brodsky and Kaspi, 2017; Tamarin and Kaspi, 2016; Teubler and Riemer, 2016) or to extratropical cyclone intensity (Seiler, 2019). Ertel (1942) defines PV as

$$q = \frac{1}{\rho}\eta \cdot \nabla\theta \tag{1}$$

where $q$ is PV, $\rho$ is the air density , $\eta$ is the absolute vorticity, $\theta$ is potential temperature, and $\nabla$ is the three-dimensional Nabla operator.

The PV inversion code used here is based on Davis and Emanuel (1991) and Davis (1992), but was strongly modified by Teubler and Riemer (2016). Nonlinear balance (Charney, 1955) assumes that the wind's irrotational component is substantially smaller than the magnitude of the nondivergent wind (e.g. Seiler, 2019; Davis, 1992). Therefore, geopotential ($\Phi$) as a function the non-divergent streamfunction ($\Psi$) and Ertel's PV can be expressed in spherical coordinates (Davis and Emanuel, 1991; Davis, 1992):

$$\nabla^2\Phi = \nabla \cdot f\nabla\Psi + \frac{2}{a^4 cos^2\phi}\left[\frac{\partial^2\Psi}{\partial\lambda^2}\frac{\partial^2\Psi}{\partial\phi^2} - \left(\frac{\partial^2\Psi}{\partial\lambda\partial\phi}\right)^2\right] \tag{2}$$

and

$$q = \frac{g\kappa\pi}{p}\left[(f+\nabla^2\Psi)\frac{\partial^2\Phi}{\partial\pi^2} - \frac{1}{a^2\cos^2\phi}\frac{\partial^2\Psi}{\partial\lambda\partial\pi}\frac{\partial^2\Phi}{\partial\lambda\partial\pi} - \frac{1}{a^2}\frac{\partial^2\Psi}{\partial\phi\partial\pi}\frac{\partial^2\Phi}{\partial\phi\partial\pi}\right] \tag{3}$$

respectively, where $\Phi$ is the geopotential, $\Psi$ is the non-divergent streamfunction, $\lambda$ is longitude, $\phi$ is latitude, $a$ is the radius of the earth, $\kappa = R/C_p$ is the Poisson constant, $f$ is the Coriolis Parameter, $p$ is pressure, and $\pi = (p/p_0)^\kappa$ is the Exner function.

Solving 2 and 3 for the unknowns $\Phi$ and $\Psi$, given $q$ on a limited domain conduces to the full PV inversion (Davis and Emanuel, 1991). Regarding boundary conditions, $\Phi$ and $\Psi$ are prescribed on the lateral domain boundaries (Dirichlet boundary




conditions) and their vertical derivatives on the horizontal boundaries (Neumann boundary conditions). On the lateral bound-
aries, the observed geopotential is employed as the boundary condition for $\Phi$ and $\frac{\partial \Phi}{\partial \pi} = -\theta$ is used at the top and bottom of the
domain (for more details see Davis, 1992).

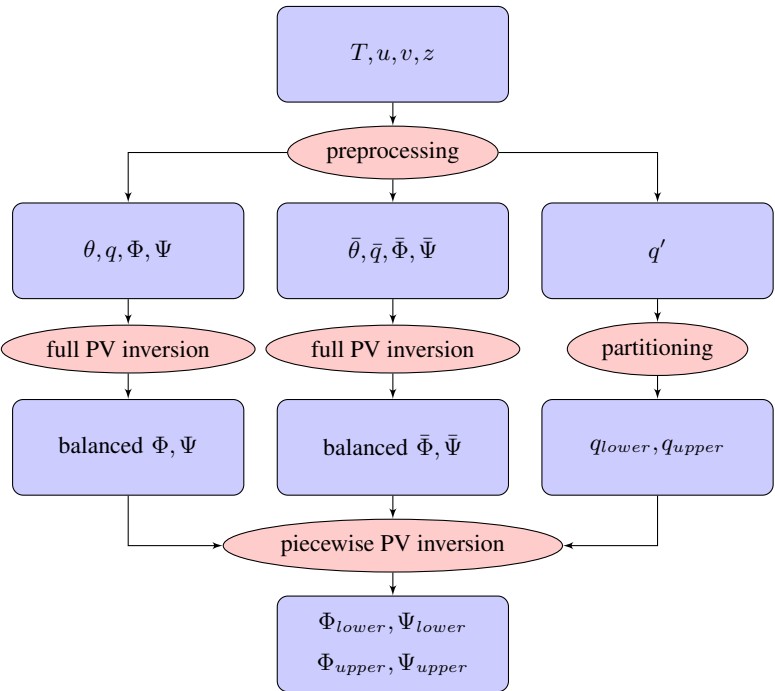

**Figure 1.** Algorithm for computing the piecewise potential vorticity inversion. Rectangles indicate variables, including temperature $T$, zonal
wind speed $u$, meridional wind speed $v$, geopotential height $z$, potential vorticity $q$, geopotential $\Phi$, and streamfunction $\Psi$, and ovals show
steps of the calculation. Adapted from Seiler (2019)

At each grid point, PV anomalies are calculated as deviations from the PV background, defined here as a running mean over
a 30 days time slice. A full PV inversion is computed for instantaneous values ($q$) and for 30-day mean values ($\bar{q}$), conducting

to instantaneous balanced $\Phi$ and $\Psi$, and mean balanced $\bar{\Phi}$ and $\bar{\Psi}$.

The PV inversion is performed for the time of maximum intensity for each cyclone track and in a three-dimensional box
bounded by the 900 and 100 hPa pressure surfaces, with a horizontal dimension of 30° longitude, 15° latitude around the
cyclone center of interest. Furthermore, PV anomalies $q'$ in upper- and lower-tropospheric layers are considered separately
from each other (right column in Fig. 1). Note that in the lower layer contribution, the PV anomalies and potential temperature

at the boundary are separated, but the upper layer contribution contains both, PV anomalies and temperature at the upper
boundary. The separation level between these anomalies is chosen as 600 hPa, which is consistent with the transition of the
shape of the PV features from lower to higher levels (see Supplementary Fig. S2). Finally, also the temperature anomaly on the
lower boundary (875 hPa) is inverted separately. See figure 1 for a general overview of the PPVI algorithm.





The CESM-LENS output is provided on hybrid sigma-pressure coordinates and, for the PPVI, is interpolated to equidistant
isobaric levels ($\Delta P = 50 hPa$) between 1000 and 50 hPa. Furthermore, in regions of high topography, the data are extrapolated
below the ground. Following (Davis and Emanuel, 1991), temperature is extrapolated using a constant lapse rate (moist), then
the geopotential is obtained by using the hydrostatic equation and state equation. Finally, the missing wind components (u,v)
below the ground are filled by propagating the last value available downward.

Because of the computational effort, the piecewise PV inversion is not performed for all cyclones mentioned above (the 10%
strongest), but only for the 1% strongest cyclones, which are denoted as extreme cyclones in the following.

## 4   Results

### 4.1   Storm tracks

In order to provide an overview of the simulated cyclone climatology and put the following, more specific results in context,
this section describes the climatology of cyclones in the CESM-LENS simulations in comparison with ERA-Interim data.
Subsequently, projected changes in winter storm tracks and cyclone intensity in a warmer climate at the end of the century over
the North Atlantic are presented.

Figure 2a shows the storm track density from ERA-Interim for the extended winter season (October-March). A region of
high cyclone frequency extends from the east coast of the USA to northern Europe. The cyclone density maximum is located to
the south of Greenland. The Mediterranean region is another area with enhanced cyclone activity. This storm track distribution
is similar to previous studies (e.g. Wernli and Schwierz, 2006; Ulbrich et al., 2009; Zappa et al., 2013).

The CESM-LENS model bias for the winter months is shown in figure 2b. In general, in the North Atlantic basin, the model
tends to produce more cyclones over the USA East Coast, between Iceland and Norway, around Greenland, Southern Spain and
parts of Eastern Europe. However, there is a predominant underestimation of cyclone frequencies over the ocean. These results
are again similar to previous studies based on the CESM-LENS model (e.g. Day et al., 2018; Raible et al., 2018). Nevertheless,
the CESM-LENS storm track is not oriented too zonally over the North Atlantic, which is in contrast to many CMIP5 climate
models (see Zappa et al., 2013). For the interpretation of this bias, it is important to keep in mind that this analysis also includes
weak cyclones, which may cause noisy results in some regions. This is particularly evident for the summer season (not shown).

Aggregated over the entire North Atlantic basin, CESM-LENS reproduces cyclone frequencies and lifetimes fairly well
(Fig. 2c), in particular of those cyclones with a lifetime of six days or more. The number of shorter-lived cyclones is slightly
underestimated compared to ERA-Interim.

The cyclone activity response in a warmer climate at the end of the century [future (2091-2100) - present (1990-2000)]
is particularly variable over the North Atlantic region (Fig. 3a). Despite this variability, there are some evident changes. The
cyclone frequency decreases over the main storm track region, the USA East Coast, southern Greenland, and the Mediterranean
region. Figure 3b indicates that this decrease is mainly related to cyclones with a short life of 2 days or less. A similar reduction
in cyclone frequencies has been reported in other studies using the CESM-LENS Model (e.g. Raible et al., 2018; Day et al.,
2018) and other CMIP5 models (Zappa et al., 2013). Figure 3a also indicates more substantial cyclone frequency changes



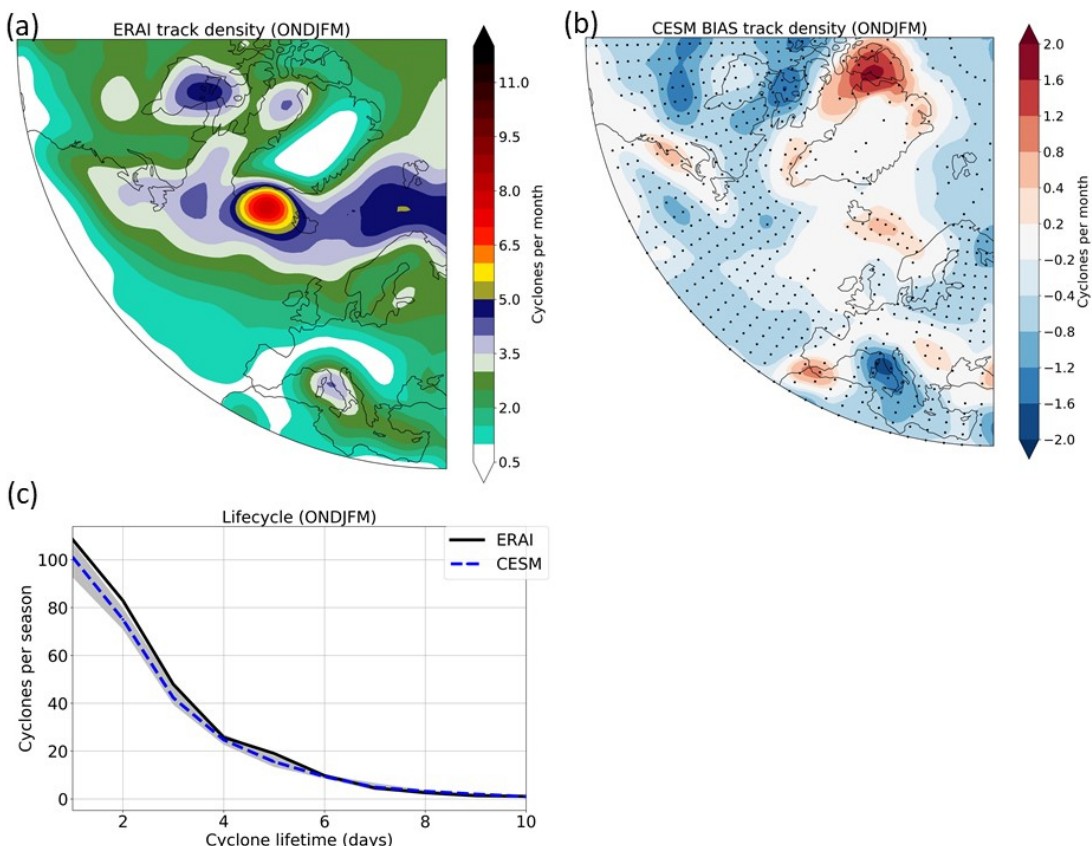

**Figure 2.** a) Cyclone track density (number of cyclones per month per unit area) in ERA-Interim (1979-2010) over the North Atlantic for the extended winter season. b) Mean track density bias of CESM-LENS (1990-2000) relative to ERA-Interim (1979-2010). Black dots in (b) denote regions of ensemble agreement on the sign of bias, i.e., more than 80% of the ensemble members indicate a bias of the same sign. c) Statistical cyclone count as a function of cyclone lifetime in ERA-Interim (1979-2010) and CESM-LENS (1990-2000) for the extended winter season in the North Atlantic region. Ensemble mean (blue dashed line) and spread of ensemble members 1-10 (grey colour shading) are shown.

at higher latitudes around Greenland, in particular an increase in cyclone frequency to the west and north of Greenland. A slight increase is also found north of the United Kingdom and west of the Scandinavian peninsula. Zappa et al. (2013) show a more evident increase over the United Kingdom, based on their multi-model analysis, which is less evident in figure 3a. This difference may be due to a specific signature in CESM-LENS. Still, it might also be related to differences in the cyclone tracking scheme and to the fact that we considered the months October-March while Zappa et al. (2013) analyzed December-February.

As a measure of extratropical cyclone intensity, we use the relative vorticity at 850 hPa (RV850) averaged over a radial cap of a radius of 250 km around the location of the sea level pressure minimum for each time step during cyclone lifetime. The





spatial pattern of projected future changes in cyclone intensity is shown in figure 3c. Extratropical cyclones are projected to become less intense over the main storm track and the Mediterranean region but more intense over the Norwegian Sea, northern Scandinavia and to the west and north of Greenland. Also, over Central Europe, there is an area of a slight increase in cyclone intensity. However, less than 8 of 10 ensemble members agree on the sign of this change.

Figure 3d shows the frequency distribution of cyclone intensity for the entire North Atlantic regions in present-day and 210  future climate. The distributions are similar (no substantial change in cyclone intensity) and for most intensities, the frequency slightly decreases between present-day and future climate. There is only a subtle increase in the frequency of intense cyclones. The future-climate 90th percentile of cyclone intensity is close to its present-day value, but the 99th percentile increases in the future. The presence of stronger winter cyclones in a warming climate has been suggested by Zappa et al. (2013); however, this result is less evident in our analysis.

We also explore projected changes in the maximum wind speed related to extratropical cyclones. The maximum wind speed at 850 hPa is found in a radius of 500 km around the location of the sea level pressure minimum for each time step during cyclone lifetime. Figures 3e and 3f show future changes in the spatial pattern and frequency distribution of the cyclone-associated maximum wind speed. The spatial patterns are similar to the changes in cyclone intensity (see again Fig. 3c) and generally tend to follow the cyclone track response, with an increase in cyclone intensity and maximum winds in regions 220  where the cyclone frequency is projected to increase. However, increases in maximum wind speed (e.g., over central Europe) are typically more pronounced than the changes in cyclone intensity. An increase in maximum wind speed of more than 10% is found north of the United Kingdom.

In summary, this section shows that the cyclone frequency biases in CESM-LENS are relatively small in the main storm track region and are mainly due to short-lived cyclones. Future projections of cyclone frequencies mostly indicate a decrease 225  over the ocean and the Mediterranean region, consistent with previous studies. Finally, cyclone intensities are not projected to change substantially overall, and the spatial variations in intensity changes are similar to frequency changes. This lack of major cyclone intensity changes is consistent with previous studies (Day et al., 2018; Raible et al., 2018; Zappa et al., 2013; Ulbrich et al., 2009; Pfahl et al., 2015) and most likely due to the competition between different thermodynamic and dynamic factors, as outlined in Sect. 1.

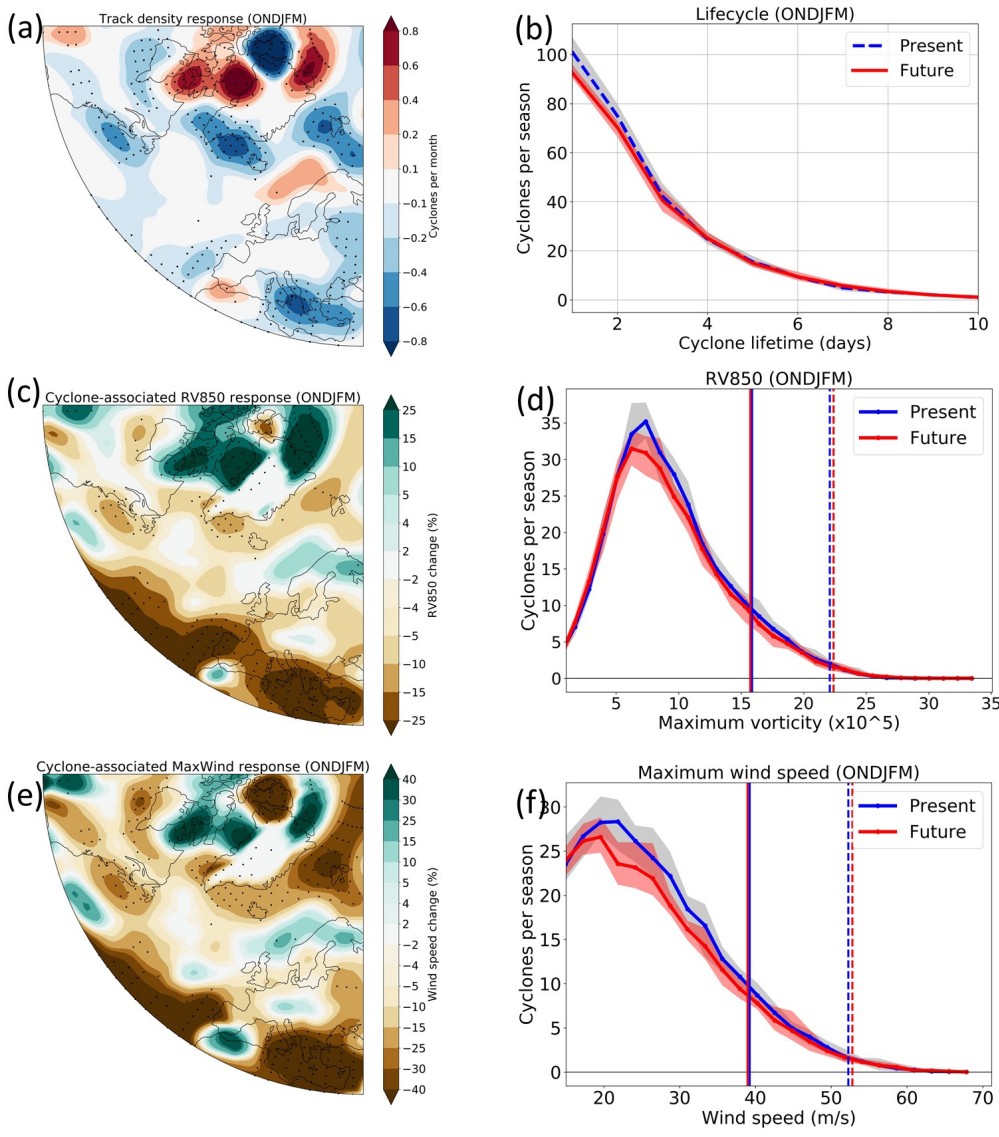

**Figure 3.** a) Mean track density response [future (2091-2100) - present (1990-2000)] from CESM-LENS in winter over the North Atlantic. b) Statistical cyclone count as a function of lifetime in present-day and future climate simulations. Ensemble mean (blue dashed line) and spread of ensemble members 1-10 (grey color shading) are shown. c) Cyclone intensity response (%), measured in terms of averaged RV850. d) Frequency distributions of cyclone intensity. e) Response of cyclone maximum wind speed at 850 hPa (%). f) Frequency distributions of cyclone maximum wind speed. The blue lines in d and f represent present-day climate (1990-2000) and the red lines represent future climate (2091-2100). The vertical lines show the 90th (solid) and 99th (dashed) percentiles, and the spread of ensemble members 1-10 is shaded. Black dots in (a, c, e) denote regions of ensemble agreement on the sign of change, i.e., more than 80% of the ensemble members indicate a change of the same sign.




## 4.2 Composites of intense extratropical cyclones

In the previous section, we see that there are only minor projected changes in cyclone intensity in CESM-LENS. However, even if overall intensity changes are small, there may be changes in cyclone dynamics and structure that can affect, e.g., spatial patterns of near-surface winds. Hence, we focus in more detail on the cyclone structure and dynamic properties in the following. The leading hypothesis is that changes in cyclone impacts, such as wind gusts, do not occur only because of the shift in storm tracks but also because the properties of individual cyclones may change in a warming climate. Therefore, this section aims to investigate storm-scale dynamical changes with the help of composite analysis.

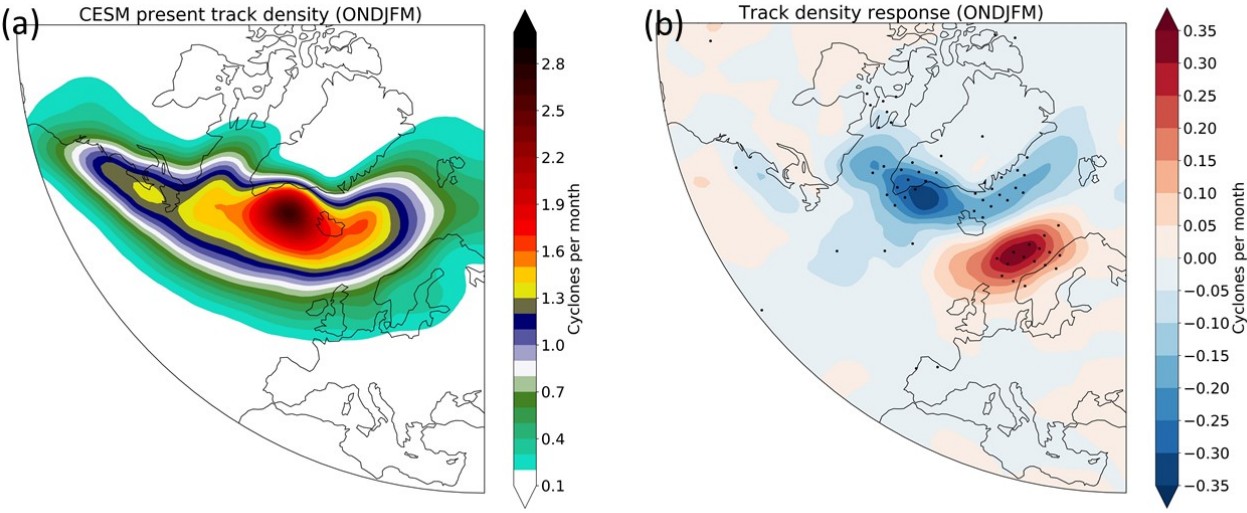

**Figure 4.** a) Mean track density (1990-2000) of intense cyclones and b) corresponding mean density response [future (2091-2100) - present (1990-2000)] from CESM-LENS in winter over the North Atlantic. Intense cyclones are defined as the 10% strongest systems in terms of maximum RV850. Black dots in b) denote regions of ensemble agreement on the sign of change, i.e., more than 80% of the ensemble members indicate a change of the same sign.

For the composite analysis, we use the 10% strongest storms over the North Atlantic in each ensemble member. In this way, the focus is laid on the main storm track region over the ocean (as shown in Fig. 4a), where the model also has the smallest biases compared to other regions such as the Mediterranean. In addition, the 10% strongest cyclones are the most relevant ones in terms of wind impacts.

Figure 4b shows the future change in intense cyclone frequency. A bipolar pattern dominates the response in track density. An increase in intense cyclone frequency is projected over the northeast Atlantic region. The increase spreads over the northern United Kingdom and the western coast of Norway. In contrast, there is a reduction in track density to the south and east of Greenland. These results are again similar to the strong cyclone response found by Zappa et al. (2013), who defined strong cyclones as those exceeding a threshold in the maximum along-track wind speed at 850 hPa. However, 4b indicates a more





evident increase than the results shown by Zappa et al. (2013) in cyclone frequency in the eastern North Atlantic. Figure 4b also confirms that the notable changes in the frequency of all cyclones in figure 3a over west Greenland and high latitudes are due to relative weak storms.

Figure 5 shows the composites of relative vorticity at 850 hPa (RV850) and SLP (Fig. 5a), wind speed at lower levels (850 hPa, Fig. 5c) and wind speed at upper levels (250 hPa, 5e) for intense cyclones in present-day climate (1990-2000) at the time of their maximum intensity. RV850 (Fig. 5a) is largest in a region of approximately $\pm\ 2.5^o$ around the cyclone center. The horizontal gradient of RV850 is larger upstream than downstream of the cyclone center. Therefore, high values of RV850 (values above $4\mathrm{x}10^{-5}s^{-1}$) cover a bigger area to the east of the SLP minimum, in the region of the warm front.

The wind speed at 850 hPa (Fig. 5c) shows a well-defined cyclonic circulation with equatorward flow upstream and poleward flow downstream of the cyclone center. The regional maximum in wind speed at 850 hPa is located to the southeast of the cyclone center, over the warm sector, where the highest values are above 26 m/s. Similar low-level wind patterns with the strongest winds in a region from southwest to east of the cyclone center have been found in previous studies (Slater et al., 2017, 2015). The wind speed maximum in figure 5c may be related to the low-level jet that typically occurs ahead of the cold 260 front. This low-level jet and the associated wind shear can result in strong wind gusts at the surface (Lackmann, 2002). Note, however, that the compositing method may average out some of the air-stream related features. For instance, some storms may have the wind maximum behind the cold front associated with the CCB, others in the low-level jet region ahead of the front, and averaging then produces a broader maximum covering the entire area.

The upper-level wind composite (Fig. 5e) shows the jet stream and its local maximum (jet streak) to the south of the cyclone 265 center. The jet's wind direction is eastward to the south and poleward to the east of the cyclone center. The poleward flow to the east enhances the poleward motion of the low-level cyclone, which can contribute to cyclone intensification when it crosses the upper-level jet axis (Rivière et al., 2013; Tamarin and Kaspi, 2017). At their time of maximum intensity, the most intense cyclones are typically located near the left exit region of the upper-level jet streak, as also evident from figure 5e, in an area of strong quasi-geostrophic forcing (Deveson et al., 2002; Barnes and Colman, 1993).


Changes in the composite patterns shown in figure 5b,d,f are indicative of projected future changes in cyclone structure. The relative vorticity changes (Fig. 5b) have an intricate spatial pattern, with a RV850 reduction near the cyclone center and RV850 increases to the northeast and southwest of the SLP minimum. This may indicate enhanced relative vorticity values near the cold and warm fronts of intense cyclones in a warmer climate, as also suggested by Sinclair et al. (2020), who found a similar 275 spatial pattern in aquaplanet simulations.

Figure 5d shows the wind speed changes at 850 hPa. A slight decrease in wind speed is projected north and northeast of the cyclone center, but in most other regions, wind velocities increase. The most significant increase emerges to the southeast of the center. This is related to a broadening of the footprint of strong winds (see again Fig. 5c) in southeastward direction, further into the cyclones' warm sector.



**Figure 5.** Present-day composites for intense cyclones of a) relative vorticity, c) wind speed at 850 hPa and e) wind speed at 250 hPa, and b, d, f) their future change for winter in the North Atlantic region. Mean SLP (hPa) is overlaid as black contour lines in a). The present-day mean of each field is overlaid as black contour lines in b, d and f. Green dots in b, d, and f denote regions of ensemble agreement on the sign of change, i.e., more than 80% of the ensemble members indicate a future change of the same sign. The composites are shown at the time of maximum intensity (time=0).

The wind speed changes at 250 hPa are shown in figure 5f. An increase of the wind velocity can be identified to the south and downstream of the cyclone center, linked to a more robust eastward flow and an enhanced jet stream. This increase is




consistent with the mean response of the jet stream to enhanced upper-tropospheric meridional temperature gradients (Grise and Polvani, 2014; Shaw et al., 2016). Note that, at both vertical levels, the wind increases are more robust than the decreases across the ensemble members (see the green dots in Fig. 5d and 5f).

In the following, in order to better understand the changes in cyclone dynamics that are related to these altered wind patterns, we investigate composite potential vorticity anomalies and their projected future changes.

### 4.3 PV analysis

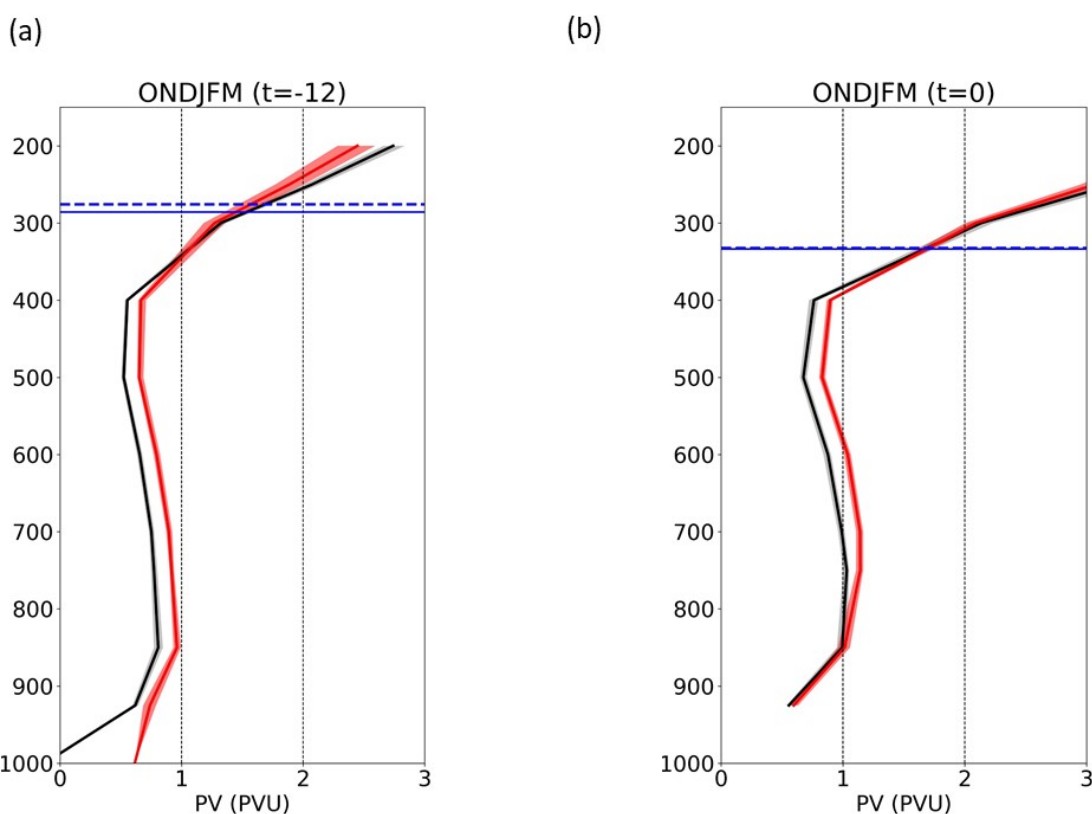

**Figure 6.** Vertical profiles of PV anomalies associated with intense extratropical cyclones in the North Atlantic region during winter for present-day climate (1990-2000), shown as black lines, and future climate (2091-2100), shown as red lines. Anomalies are calculated with respect to a local PV climatology and averaged in a radius of 2.5° around the cyclone center. The solid blue lines indicate the 2-PVU tropopause in the presence of cyclones for present-day (solid line) and future (dashed line) climate. Composite profiles are shown 12 hours before (a) and at the time of maximum intensity (b).

To first quantify simulated PV changes averaged over the cyclone area, figure 6 shows the PV anomaly profiles for intense cyclones in the North Atlantic region. At the time of maximum intensity (t=0, figure 6b), there are prominent positive PV





anomalies in the lower and upper troposphere, as typically associated with intense extratropical cyclones (Grams et al., 2011; Čampa and Wernli, 2012; Pfahl et al., 2015; Büeler and Pfahl, 2017). The lower-tropospheric PV anomaly is primarily created by diabatic processes, such as latent heat released during cloud formation (Wernli and Davies, 1997; Ahmadi-Givi et al., 2004; Büeler and Pfahl, 2017). The upper-tropospheric PV anomaly is associated with adiabatic PV advection from the stratosphere and partly influenced by the cyclonic wind field linked to the lower-tropospheric PV anomaly (Hoskins et al., 1985; Wernli

et al., 2002). Figure 6a shows the PV anomaly profile 12 hours before the time of maximum intensity when both PV anomalies are still weaker and the low-level maximum is located at a higher pressure. As indicated by the blue lines in figure 6, the growth of the upper-tropospheric PV anomaly between t=-12 and t=0 goes along with a lowering of the dynamical tropopause.

   As a response to climate warming, at the time of maximum intensity (t=0, Fig. 6b), there is a general increase of the lower and mid-tropospheric PV anomaly and a slight decrease in the upper-tropospheric PV anomaly. Note that the limited change

in the upper-level anomaly is related to the relatively small averaging radius of 2.5°. Changes in the upper-level PV structure will become more evident in the composites discussed below. The amplification of the lower tropospheric PV anomaly is more robust in the layer between 850 and 600 hPa. These structural changes are similar to the previous finding of Büeler and Pfahl (2019) for intense cyclones based on idealized model simulations. For instance, also in these idealized simulations, the lower tropospheric PV maximum extends more into the middle troposphere in warmer climates. However, the magnitude of PV

changes in the Büeler and Pfahl (2019) experiments are more prominent than in the CESM-LENS simulations presented here, which is mainly due to the fact that the range of climate warming is much more extensive in the idealized simulations.

   The positive PV anomaly in the lowest part of the troposphere (below 850 hPa) increases more in the hours before maximum intensity (Fig. 6a). This increase is likely related to a more considerable influence of cloud formation and precipitation processes prior to the maximum intensity (Pfahl and Sprenger, 2016), which agrees with an analysis of the precipitation life cycle that

will be presented in the second part of this study.

   Finally, figure 6 indicates that the height of the tropopause during the presence of intense cyclones is projected to slightly increase, which reflects the general increase in tropospheric depth in a warming climate (O'Gorman and Schneider, 2008; Pfahl et al., 2015). This increase is more evident 12 hours before the maximum intensity.

In the following, the horizontal structure of cyclone-related PV anomalies is also analyzed with the help of composites. Note again that a PV climatology specific to the respective climate (present-day vs future) has been subtracted from the full PV fields to obtain these anomalies. As shown in figure 7a, there is a clear, diabatically created maximum in the PV anomaly at 700 hPa near the cyclone center, which extends downstream in the region of the warm front. A secondary maximum can be observed along the north-western flank of the composite region. The second maximum is likely related to frictional PV generation over

the land masses such as Greenland and North America that are located to the north-west of the North Atlantic storm track (see again Fig. 4a).

   The upper-tropospheric PV anomaly shown in figure 7c shows a dipole pattern, with high PV values associates with an upstream trough in the western part of the composite and a maximum south-west of the cyclone center. A clear signature of cyclonic wave breaking can be observed, which is also linked to the formation of the jet streak (see again Fig. 5e). This PV

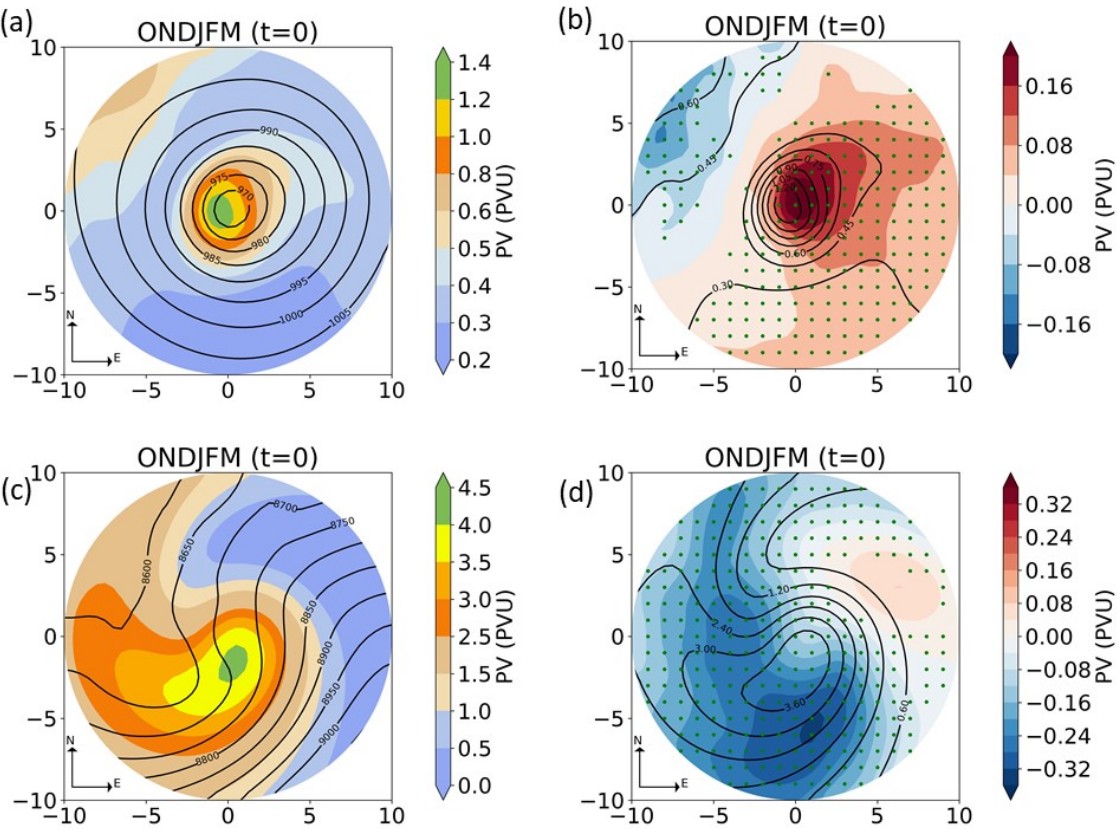

**Figure 7.** Present-day composites for intense cyclones of a) PV at 700 hPa and c) PV at 250 hPa and b,d) their future change for winter in the North Atlantic region. Mean SLP (hPa) is overlaid as black contour lines in a) and geopotential height (m) is overlaid as black contour lines in c. The present-day mean of each field is overlaid as black contour lines in b and d. Green dots in b and d denote regions of ensemble agreement on the sign of change, i.e., more than 80% of the ensemble members indicate a future change of the same sign. The composites are shown at the time of maximum intensity (time=0).

anomaly pattern is typical for extratropical cyclones during their mature stage of development (Pinto et al., 2014; Houze Jr, 2014; Michaelis et al., 2017) . Also, it is a signature of the deepening processes of the upper troposphere, which lead eventually to the "treble clef" Upper level PV (UPV) distribution characteristic of a warm-occluded structure in the lower troposphere (Martin, 1998).

Future changes of lower-tropospheric composite PV anomalies are shown in figure 7b. Positive PV anomalies are projected to increase in the region of the present-day maximum, near the cyclone center, but also downstream in the region of the warm front and generally in the area of the cyclone's warm sector. This increase is consistent with an increase in precipitation over the warm sector (as will be analyzed in more detail in another study) and with the overall rise of atmospheric moisture content, (cyclone) precipitation and thus latent heat release in a warmer climate (Schneider et al., 2010; Yettella and Kay, 2017). Similar





increases in lower-tropospheric PV have been found in other studies (e.g. Marciano et al., 2015; Michaelis et al., 2017; Zhang
and Colle, 2018; Sinclair et al., 2020) and have been directly attributed to enhanced latent heat release in idealized simulations
(Büeler and Pfahl, 2019). The increase in lower-tropospheric PV over the warm sector corresponds with an increase in relative
vorticity (cf. Fig. 5b) in some areas, such as near the warm front, but not everywhere, pointing towards a complex relationship
between PV and wind changes that will be further discussed below. Finally, PV at 700 hPa is projected to decrease in the region
of the secondary maximum at the northwestern edge of the composite. This might be related to the eastward shift of the intense
cyclone tracks in the northeastern Atlantic (see again Fig. 4b), which brings them further away from Greenland's topography.

The upper-level PV anomaly is projected to decrease in most regions, but in particular upstream and south of the cyclone
center (Fig. 7d). The most substantial reduction occurs along the southern flank of the PV maximum, pointing to a decreased
wave breaking. A small PV increase is found downstream of the cyclone center, which is, however, hardly consistent across
the ensemble member. Again, a similar reduction of upper-tropospheric PV has been found in previous studies (e.g. Zhang and
Colle, 2018). Michaelis et al. (2017) proposed that this decrease may be partly explained by increased latent heating that goes
along with amplified PV erosion above the heating maximum. In addition, the slight increase in southerly wind velocities in
upper levels over the warm sector (see again Fig. 5f) may go along with enhanced advection of low-PV air masses from lower
latitudes. This complex interplay between PV advection and diabatic tendencies (see also Brennan et al., 2008; Madonna et al.,
2014) will be further investigated in the second part of this study.

The enhanced diabatic heating in a warmer and therefore more humid climate may thus influence PV anomalies and cyclone
dynamics at both lower and upper levels. At low levels, it generates sizeable positive PV anomalies that directly contribute to
cyclone intensification.

Increased diabatic PV production in the region of the cold front my strengthen the low-level jet (Michaelis et al., 2017) (Fig.
5d), which is, however, difficult to detect in our PV composites as the cold front positions of different cyclones are not aligned.
At upper levels, amplified LH can contribute to PV reduction and locally steepen the horizontal PV gradient downstream of the
positive PV anomaly (in the region of the strongest reduction in figure 7d). Changes in upper PV distribution goes along with
enhanced wind velocities in the jet streak (Bluestein, 1992) (Fig. 5f), which again affect the PV distribution through horizontal
transport. This qualitative discussion will be completed by quantitative PPVI results below.

Due to the relatively high computational effort, PPVI is not performed for all intense cyclones, but only for the 1% strongest
cyclones in the 10 ensemble members are analysed, which are referred to as extreme cyclones here. In this way, the focus is
laid even more on the strongest storms with potentially largest impacts in terms of near-surface wind velocities. Before we
show the PPVI results, we thus briefly discuss composite PV and wind changes associated with these extreme cyclones.

In the lower troposphere, the future changes in PV and wind composites of extreme cyclones (Fig. 8a,b) reproduce the main
features of the corresponding composite changes of intense cyclones (cf. Figs. 7b and 5d), in particular regarding the regions
of PV and wind speed increases that are consistent across ensemble members. Nevertheless, differences between extreme and
intense cyclones are found in the magnitude of changes. For example, the wind magnitude increases by up to 1.2 m/s for
intense cyclones, but by up to 2.5 m/s for extreme cyclones. In addition, the region of strong wind increase reaches further to



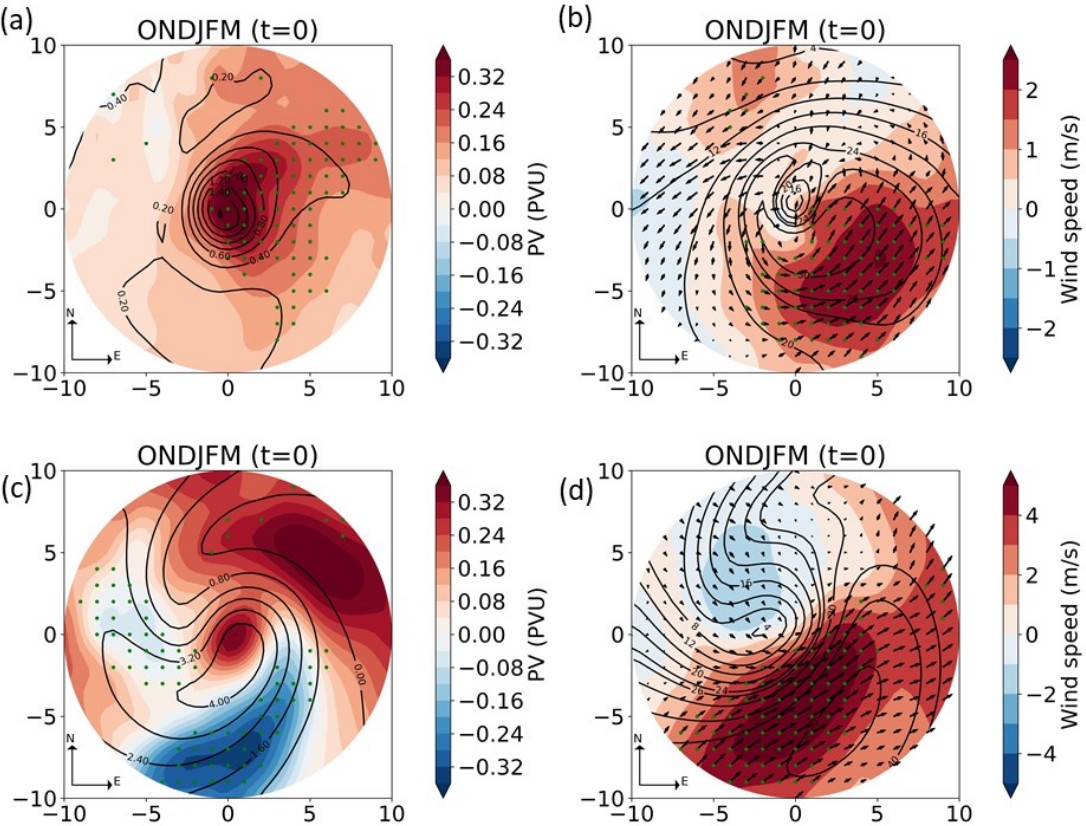

**Figure 8.** Future change in composites for extreme cyclones of a) PV at 700 hPa, b) wind speed at 850 hPa, c) PV at 250 hPa and d) wind speed at 250 hPa for winter in the North Atlantic region. The present-day mean of each field is overlaid as black contour lines. Green dots denote regions of ensemble agreement on the sign of change, i.e., more than 80% of the ensemble members indicate a future change of the same sign. The composites are shown at the time of maximum intensity (time=0). Extreme cyclones are defined as the 1% strongest systems in terms of maximum RV850.

the southeast for extreme cyclones, i.e., the wind footprint is even further enlarged, and there is a stronger poleward component
in the wind change for extreme cyclones. Also, the maximum magnitude of (diabatic) PV changes is about twice as large for the extreme as for the intense cyclones. These results further support the hypothesis that the strengthening of the low-level winds can be associated with diabatic potential vorticity increase: a larger amplification of diabatic PV generation in extreme cyclones goes along with a larger increase also in wind speed.

In contrast to the lower troposphere, upper-level PV changes for extreme cyclones differ more substantially from the intense
cyclone changes. For instance, figure 8c indicates an upper-level PV increase near the cyclone center, where a decrease is projected for intense cyclones (Fig. 7d). However, this increase is not robust across ensemble members. A PV reduction to the south and west of the cyclone center, on the other hand, is common to both extreme and intense cyclones. Upper-level





wind speed changes for extreme cyclones (Fig. 8d) again follow a similar spatial pattern as for intense cyclones (Fig. 5f) with amplified magnitude. Similar to the lower levels, the wind change in the region of the main jet has a more pronounced poleward
component in extreme cyclones.

All together, this comparison indicates that, at lower levels and with regard to upper-level winds, changes in extreme cyclones are qualitatively similar to changes in intense cyclones. There are some differences in the behaviour of the two cyclone classes in terms of projected upper-level PV changes, which can, however, be partially attributed to natural climate variability (as they are not robust across the ensemble of model simulations).

**4.4 PPVI to extratropical cyclone extremes**

A piecewise inversion is performed to quantify the relative contributions of upper and lower level PV anomalies to projected future wind speed changes in extreme cyclones and thus complement the previous qualitative discussion. The usefulness of such an inversion method to explain wind changes has been demonstrated in previous studies (e.g. Tochimoto and Niino, 2016; Tamarin-Brodsky and Kaspi, 2017).

As described in Sect. 3.3, the PPVI separates the troposphere into an upper and a lower layer, and the effect of PV anomalies in each layer is separately evaluated. Figure 9 shows PV composites averaged over these layers. The spatial distribution shows a clear difference between the upper and lower layers. The lower layer is characterized by a localized PV anomaly in the region of the cyclone center, similar to the PV anomaly composite at 700 hPa for intense cyclones (see again Fig. 7a). In contrast, the average upper-layer anomaly has a broader maximum southwest of the center and a cyclonic wave-breaking structure,
reminiscent of the intense cyclone anomaly at 250 hPa (Fig. 7c). Thus, the chosen separation level at 600 hPa appears to work well in separating these different structures (for more details, see Supplementary Fig. S2). Figure 9b,d also shows the future changes of the layer-mean PV anomaly composites. These projected changes are again consistent with our previous findings for specific levels (see Fig. 8). The lower-tropospheric anomaly increases mostly near the cyclone center and in the region of the fronts. In the upper layer, there is a reduction of PV to the south of the cyclone center and a PV increase near the center.
Figure 9e shows the temperature anomalies at the lower boundary in present-day climate. A predominant positive anomaly is found downstream and a negative anomaly upstream of the cyclone center. Future changes (Fig. 9f) are characterized by a temperature anomaly reduction to the north of the cyclone center and two regions of increased temperature, to the west and southeast of the cyclone center. The latter (the most significant increase) covers most of the warm sector. Figure 8b shows a southward wind response to the north of the cyclone center as the climate warms and stronger south-westerlies in the warm
sector. The temperature anomalies in Fig. 9f can thus be related to amplified cold air advection north and amplified warm air advection southeast of the cyclone center.

Figure 10 shows the results of the PPVI for wind speed composites at 850 hPa in present-day climate (upper row) and their projected future change (lower row). The balanced wind composite (Fig. 10a) is obtained from inverting the complete PV distribution in the specified domain (see again Sect. 3.3 for more details). It is broadly similar to the full wind composite
(present-day composites for extremes, Supplementary Fig. 3a), the wind speed increases southward and there is a band of strong winds upstream. Nevertheless, the balanced wind speed has a smaller magnitude over the cyclones' warm sector. This

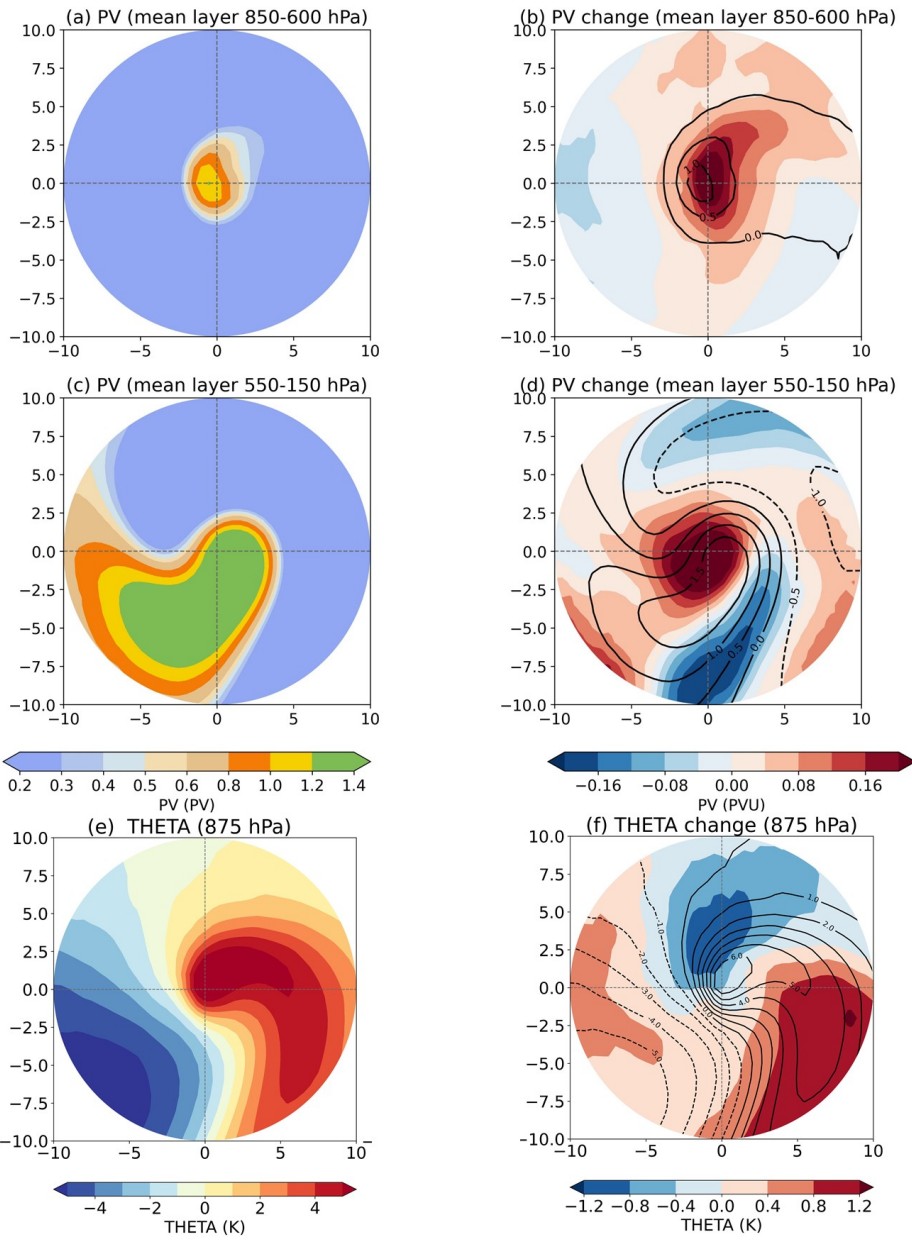

**Figure 9.** Present-day composites for extreme cyclones of PV averaged over a) the lower troposphere (850-600 hPa), c) the upper troposphere (550-150 hPa) and potential temperature at 875 hPa (lower boundary) for winter in the North Atlantic region. Future changes of the lower tropospheric PV, upper tropospheric PV and potential temperature are shown in b, d, and f respectively. The present-day mean of each field is overlaid as black contour lines in b, d and f. The composites are shown at the time of maximum intensity (time=0).

indicates that the high winds in this region, which are partly associated with the low-level jet ahead of the cold front, have an essential non-balanced component (divergent component).





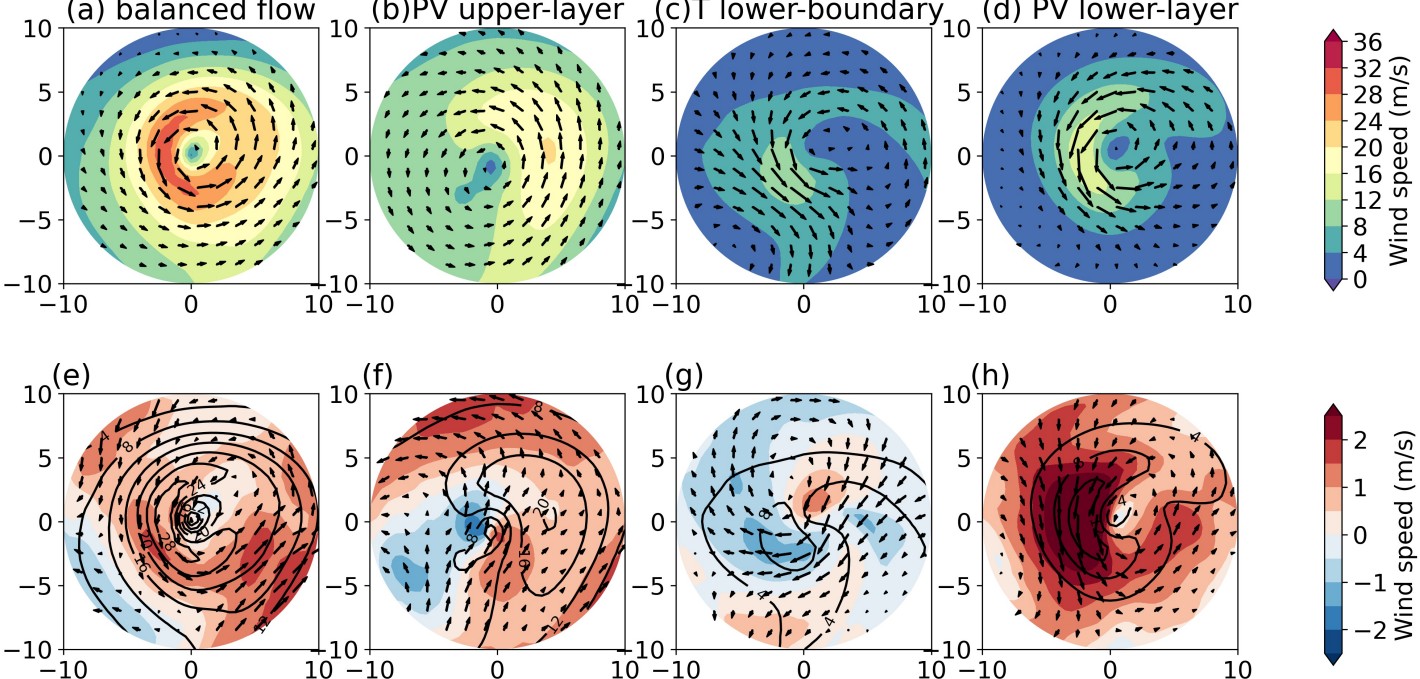

**Figure 10.** PPVI decomposition of the wind composites at 850 hPa in present-day climate (upper row) and their future change (lower row). The total balanced wind composite obtained from the full PV inversion is shown in figures a) and e). The other figures show the wind composites obtained from inverting (b, f) the upper-layer PV anomalies, (c, g) temperature anomalies at the lower boundary, and (d, h) the lower-layer PV anomalies.

Also, projected future changes in the balanced wind (Fig. 10e) reproduce changes in the full wind (Fig. 8b) fairly way. In

particular, they also show increased wind velocities and an extended wind footprint in the warm sector. PPVI allows for a decomposition of the balanced flow and its changes into contributions from different anomalies, here the upper- and lower-layer PV anomalies, temperature anomalies at the lower boundary as well as the background PV field. For the 850 hPa winds, the background only has a small contribution (see Supplementary Fig. S4).

Supplementary Fig. S5, a comparison between the balanced flow and the sum of the total contributions, demonstrates the

consistency of this decomposition and shows that both the present-day balanced flow and its projected change are similar to the sum of the individual contributions. Differences between the balanced flow and the sum of all wind components arise due to i) the imperfect knowledge of boundary conditions, ii) non-linearities associated with the separation, especially the separation of low-level temperature anomalies and low-level PV anomalies, and iii) numerical inaccuracies, mostly in calculating the Neumann boundary condition at 125 hPa, where the vertical $\theta$ gradient is very large.

The decomposition of the present-day wind composite (Fig. 10a-d) indicates that all three anomalies contribute to the cyclonic circulation. The weakest contribution comes from the temperature anomaly, whose center of circulation is also shifted





northwestward compared to the cyclone center. The upper-layer PV anomaly has overall the largest contribution, in particular for the poleward flow downstream of the cyclone center, where the horizontal PV gradients in the upper layer are also the largest (cf. Fig. 9c). Low-layer PV contributes most substantially to the equatorward flow upstream of the center, again in the

region of largest low-layer PV gradients. The critical role of upper-layer PV for the flow at 850 hPa seen in figure 10 is generally consistent with our conceptual understanding of cyclone intensification (Hoskins et al., 1985) and with previous PPVI results (Seiler, 2019).

The future change of PPVI decomposition of the wind flow at 850 hPa is discussed now. The largest contribution to the future changes in balanced flow at 850 hPa (Fig. 10e) is associated with low-layer PV changes (Fig. 10h). The increase in low-

layer PV at the cyclone center (cf. Fig. 9b), which is mainly due to enhanced latent heating (see again Sect. 4.3), is linked to amplified cyclonic circulation all around the cyclone center, but in particular to its west, where the already strong PV gradients are further enhanced. However, this amplification of the equatorward flow upstream is partly compensated by a reduced equatorward component (Fig. 10f) associated with a reduction in the upper-layer PV gradient due to a PV increase to the northwest of the present-day maximum (Fig. 9d). In contrast, in the warm sector southeast of the cyclone center, both changes in lower-

and upper-layer PV are associated with an enhanced south-westerly flow and thus add up to produce higher wind velocities and an enlarged wind footprint (see again Fig. 10e). This wind speed increase in the warm sector is slightly reduced by changes in the temperature anomaly at the lower boundary that are linked to an anomalous anti-cyclonic circulation centered northwest to the cyclone center (Fig. 10g). The anti-cyclonic circulation is located over a cold anomaly (Fig. 9f). All together, these results show that the projected enhancement of the wind footprint in the warm sector cannot be attributed to a single mechanism alone,

but results from a constructive superposition of wind changes linked to upper- and lower-tropospheric PV anomalies.

Also, for wind speed in the upper troposphere, at 250 hPa (Fig. 11), the balanced flow from the PV inversion reproduces the present-day wind composite (cf. Supplementary Fig. 3b) and its projected future change (cf. Fig. 8d) qualitatively. However, the present-day wind velocities are underestimated (Fig. 11a), and the small projected wind speed reduction northwest of the

cyclone center (which is not consistent across ensemble members) is not captured (Fig. 11d). As for the low-level wind, the PPVI yields a consistent decomposition of this balanced flow and its future change (Supplementary Fig. S6). Nevertheless, the contributions of low-layer PV and temperature anomalies at the lower boundary are small (not shown), such that only contribution of upper-layer PV and the background flow are shown in figure 11.

The present-day wind composite at 250 hPa (Fig. 11a) is a superposition of westerly background flow with a wind speed

maximum at the southern edge of the domain (Fig. 11c) and a component linked to the upper-layer PV anomaly (Fig. 11b), which is mostly poleward to the east and westward to the west of the cyclone center.

The future wind speed increase at upper levels, which is mostly found to the south and east of the cyclone center (Fig. 11d), results from a combination of the two mentioned components. First, an amplified westerly background flow (Fig. 11f), consistent with previous studies of future changes in the upper-level jet (Grise and Polvani, 2014; Shaw et al., 2016). Second,

the larger wind velocities linked to upper-layer PV changes (Fig. 11e). The latter is most prominent in a region of enhanced PV gradients associated with the dipole pattern of projected PV changes (see again Fig. 9d) southeast as well as north of the cyclone





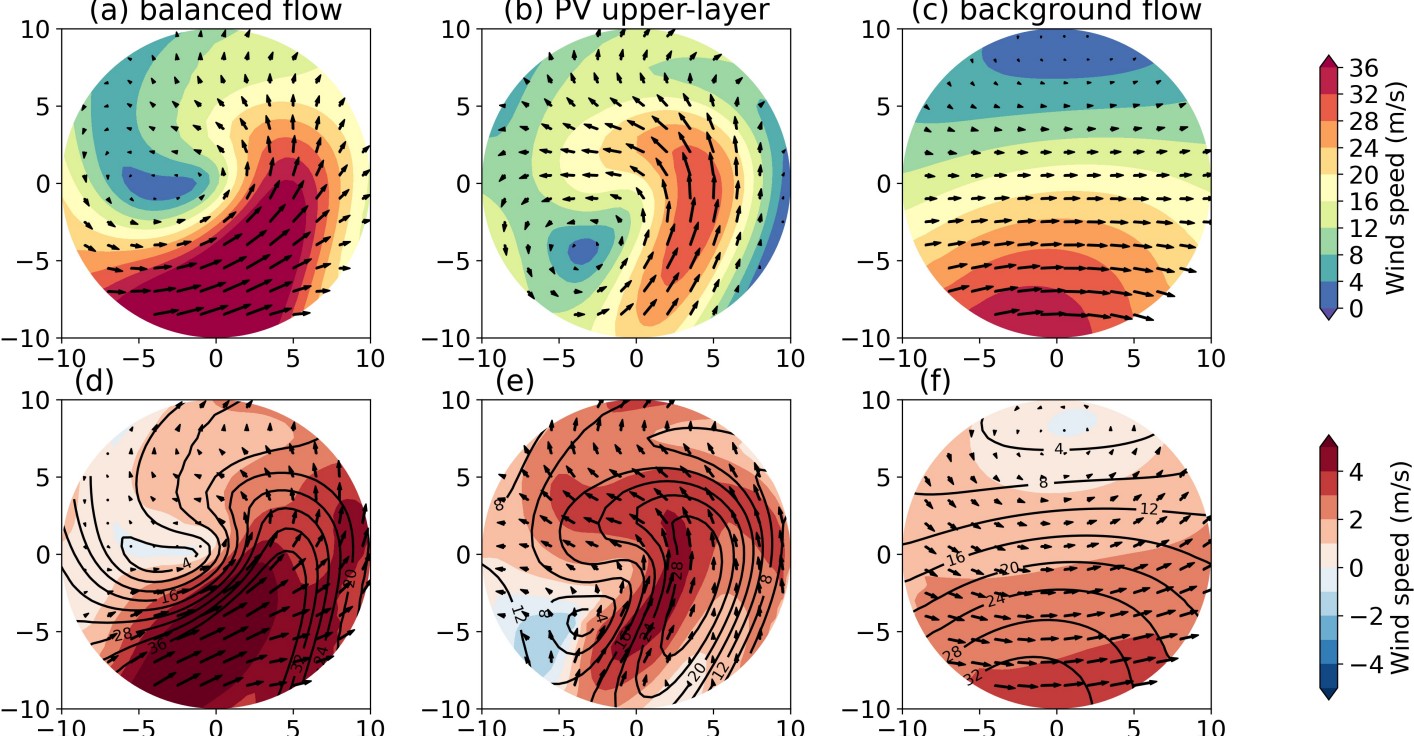

**Figure 11.** PPVI decomposition of the wind composites at 250 hPa in present-day climate (upper row) and their future change (lower row). The total balanced wind composite obtained from the full PV inversion is shown in figures a and d. The other figures show the wind composites obtained from inverting (b, e) the upper-layer PV anomalies and (c, f) the background PV.

center. Similar patterns of upper-level wind speed changes in midlatitude cyclones have been found by Tamarin-Brodsky and Kaspi (2017). The enhanced poleward flow is thought to contribute to the increased poleward propagation of cyclones in a warmer climate (Tamarin-Brodsky and Kaspi, 2017) and, as discussed in Sect. 4.3, can feedback on the PV anomaly pattern
by transporting low-PV airmasses from lower latitudes towards the cyclone center.

## 5 Conclusions

In this study, an analysis of cyclone dynamics and wind changes in a warming climate has been performed with the help of coupled climate model simulations and a potential vorticity framework. A cyclone tracking scheme as well as composite and PV inversion diagnostics have been applied to 10 CESM-LENS ensemble members. The model reproduces cyclone frequencies
over the North Atlantic well, in particular over the main storm track regions. Model biases are mostly associated with the representation of short-lived systems.





At the end of the century, projected changes in cyclone frequencies are relatively small, with a general tendency towards slight decreases in many regions. Nevertheless, for the 10% most intense cyclones, an eastward displacement of the main oceanic storm track over the eastern North Atlantic is projected, associated with an increase in cyclone track density over
northwestern Europe. Also, projected cyclone intensity changes, measured in terms of lower-tropospheric maximum relative vorticity or wind speed, are relatively small. These findings are generally consistent with previous studies (Zappa et al., 2013; Catto et al., 2019).

In spite of such small overall intensity changes, our composite analysis indicates structural changes in the typical wind patterns associated with intense North Atlantic cyclones. In particular, an increase of wind velocities in the warm sector
southeast of the cyclone center, potentially related to strengthening the low-level jet ahead of the cold front, and a southeastward broadening of the associated footprint of strong winds is projected. Together with the eastward shift of storm tracks, this may lead to increased wind hazards in western Europe, which has also been seen in other model studies (Mölter et al., 2016).

In order to better understand the dynamical mechanisms behind these wind speed changes, a PV anomaly and inversion analysis have been conducted. In agreement with many previous studies (Pfahl et al., 2015; Marciano et al., 2015; Michaelis
et al., 2017; Zhang and Colle, 2018; Sinclair et al., 2020), we find an increase in lower-tropospheric PV near the cyclone center and fronts that is most likely due to increased latent heating in a warmer and thus more humid climate (Büeler and Pfahl, 2019). According to our PPVI analysis, this amplified low-level PV is associated with enhanced cyclonic wind velocities around the cyclone center, indicating that the increased latent heating contributes to the broadening of the wind footprint in the warm sector. However, it is not the sole cause of this broadening, as also PV changes in the upper troposphere go along with an
increase of south-westerly winds in this region. More specifically, a dipole change in upper-tropospheric PV with a projected PV increase near the cyclone center and a decrease to the south and southwest are associated with enhanced upper-level PV gradients in the region ahead of the cold front and thus increased poleward flow throughout the troposphere. The projected PV reduction that is part of this dipole pattern is likely due to a combination of the advection of low-PV air by the enhanced poleward flow and potentially also enhanced diabatic PV erosion above the maximum of latent heating. In contrast to the warm
region southeast of the cyclone center, where wind changes associated with upper- and lower-layer PV changes superimpose in a constructive way, these wind changes partly compensate each other upstream of the cyclone center, where net wind speed changes are thus smaller. Note that the projected wind increase in the warm sector is robust across the different cyclone intensity classes and ensemble members analyzed here and are also seen in the balanced wind response of the PV inversion. However, the PPVI does not exactly reproduce the full wind changes in a quantitative way, and the corresponding results should thus be
interpreted with care.

In summary, the PV analysis performed in this study provides insights into the role of altered upper-tropospheric dynamics and increased latent heat release in a warmer climate for future changes in near-surface wind fields around extratropical cyclones. The projected broadening of the wind footprint southeast of the cyclone center that can be explained by a combination of these processes may have important consequences for future changes in wind hazards. In the second part of this study,
Lagrangian air stream analyses will be used to complement and expand these dynamical insights.





*Code and data availability.* The CESM-LENS data as well as the PPVI code used here are available upon request from the authors.

*Author contributions.* S.P. and E.D.-T. designed the study. E.D.-T. performed the analysis, with support of F.T. for the PPVI, produced the figures and drafted the manuscript. All authors discussed the results and edited the manuscript.

*Competing interests.* Stephan Pfahl is executive editor of WCD.

*Acknowledgements.* We are grateful to Urs Beyerle (ETH Zurich) for performing the CESM-LENS re-runs. S.P. acknowledges support by Deutsche Forschungsgemeinschaft through Grant CRC 1114 'Scaling Cascades in Complex Systems', project C06.



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
