# Peer review of "Future changes in North Atlantic winter cyclones in CESM-LENS. Part I: cyclone intensity, PV anomalies and horizontal wind speed"

_Weather and Climate Dynamics, 2021_

## Referee Comment (RC1)

**Review- Future changes in North Atlantic winter cyclones in CESM-LENS. Part I: cyclone intensity, PV anomalies and horizontal wind speed**

I have reviewed the paper the current manuscript by Edgar Dolores-Tesillos, Franziska Teubler, and Stephan Pfahl. In this manuscript the authors investigate cyclone-centered composites in an ensemble of CESM-LENS simulations. They perform a piecewise PV inversion to study the projected changes in upper and lower PV anomalies and associated winds of strong midlatitude cyclones. Overall, this is a well-written paper that presents interesting analysis. While most of the results are not really surprising (as other previous studies have already examined changes in storm-centered composites in other climate-change simulations), the detailed examination of the PPVI analysis is interesting and deserves publication, after some revision.

My detailed comments are given below

- Line 124: "15-day average"- should this be 30?

- Line 125: could the 2.5 spatial average hide some of structure at upper levels? For the full upper level PV field that should not matter, but for the upper level PV anomaly that spatial averaging around the center might involve averaging over positive (from the trough to the west) and negative (from the ridge to the east) values.

- In general, your references (when citing more than one in the same parenthesis) should appear in chronological order).

- In what ways was the PV inversion code modified by Teubler and Riemer (2016)? And do you use the modified version? Equations (2) and (3) are based on the original David and Emanuel (1991) and David (1992) papers, so this is not clear.

- Figure 1: the "lower" and "upper" decomposition should also contain a third "surface" contribution.

- Line 154: "30 days time slice"- 30 day time slice?

- Line 163: How is the separation defined? is it 600-200 hPa for the UPV, 600-875 hPa for the LPV, and 875hPa for the lower boundary? You should write this explicitly. Also, why do you take the 875 hPa as

the lower boundary for the PPVI? Can't you use the surface temperature (or T2m temperature)? You later mention that you interpolate to 1000-50hPa, so why not use the lowest level? This might influence the PPVI composites.

- Line 166: "Following (Davids and Emanuel, 1991)..."- Following Davids and Emanuel (1991),...

- Figure 3: The switching of the colormap from panel (a) to panels (c) and (e) is really confusing! Please use the same colormap, and the same color to denote the same sign (e.g., blue to denote negative values, and red to denote positive values).

- Lines 270-275: The RV850 changes (Fig.5b) essentially show a more SW-NE elongated low level feature. Could this be a signature of enhanced Anticyclonic Wave Breaking (AWB), which is projected in the future in the region?

- Figure 6: Why does PV increases everywhere in the troposphere? If these changes are due to enhanced LHR than I would expected positive PV changes below ~700-800 hPa, and negative PV changes above. This is not what you get.

- Lines 341-349 (about the upper level PV decrease): your findings about the upper level PV reduction are confusing. First, the spatial distribution of the changes (Fig.7d) are confusing- they are not similar to other studies (e.g., Michaelis 2017 see their Fig. 8g,h), who find a PV decrease to the northeast of the UPV center (opposite from what you find!). The regions to the northeast (e.g., where the low level PV increases, Fig.7b) is where I would expect to find a PV decrease at upper levels, so I am confused.
    Could this reduction of PV that you find be just a signature of the upward lifting of the troposphere and tropopause? Can you plot vertical cross sections (without the spatial averaging) of PV and PV changes? Perhaps the peak of the PV maximum at upper levels has just shifted slightly upward?

- Lines 354-359: Again, not clear to me. The diabatic heating from LHR usually tends to amplify the upper level ridge downstream of the positive PV anomaly, and this is not what you get.

- Line 414: fairly way–> fairly well

- Figure 10a: I find it confusing that the strongest meridional winds are to the west (i.e., southward), as I would expect, due to the induced winds from the UPV, to find the strongest v' to the east (i.e., poleward). Do you get the same results in the 850 hPa composites for the ERAI reanalysis data?

- Figure 10d: Why is the wind velocity induced by the low level PV alone not more circular, as one would expect from the low level roughly circular PV anomaly?

- Figure 11: If you perform the PV inversion but not show the contributions from other layers, then this is not very meaningful (panels b and c are then just like taking a mean and perturbations). I think you should at least add the low level inversions to the SI.

---

## Author Comment (AC1)

**Answer to Reviewer 1**

I have reviewed the paper the current manuscript by Edgar Dolores-Tesillos, Franziska Teubler, and Stephan Pfahl. In this manuscript the authors investigate cyclone-centered composites in an ensemble of CESM-LENS simulations. They perform a piecewise PV inversion to study the projected changes in upper and lower PV anomalies and associated winds of strong midlatitude cyclones. Overall, this is a well-written paper that presents interesting analysis. While most of the results are not really surprising (as other previous studies have already examined changes in storm-centered composites in other climate-change simulations), the detailed examination of the PPVI analysis is interesting and deserves publication, after some revision.

We appreciate and thank the reviewer for reading the manuscript and for the constructive comments. In the following we respond to all comments point by point. The line numbers and figure references in the reviewer's comments refer to the original manuscript. The reviewer's comments are in black and our response are in blue.

- Line 124: "15-day average"- should this be 30?

We will modify this sentence to: 30-day average

- Line 125: could the 2.5 spatial average hide some of structure at upper levels? For the full upper level PV field that should not matter, but for the upper level PV anomaly that spatial averaging around the center might involve averaging over positive (from the trough to the west) and negative (from the ridge to the east) values.

Yes, the average tends to smooth the anomalies. We mention in line 300 that the changes in the upper level PV structure are more evident in the composites than in the PV profiles.

Nevertheless, we would like to keep the PV profiles to compare our results with previous studies that also show PV profiles (e.g., Campa and Wernli, 2012, Büeler and Pfahl, 2017).

- In general, your references (when citing more than one in the same parenthesis) should appear in chronological order).

We will order the references chronologically in the text.

- In what ways was the PV inversion code modified by Teubler and Riemer (2016)? And do you use the modified version? Equations (2) and (3) are based on the original David and Emanuel (1991) and David (1992) papers, so this is not clear.

Thank you for your question. You are right, our point was not made clear enough. The basic equations have not been modified. There has been mainly modifications

in the code to improve the stability of the inversion, e.g. Teubler and Riemer 2016 replaced the linear solver from a SOR (successive over-relaxation) to a state-of-the art solver (fgmres - **flexibel generalized minimal residual method**). We removed that bit of information and reformulated that sentence accordingly to:

Here we use piecewise PV inversion based on nonlinear balance (Davis and Emanuel 1991, Davis 1992).

-Figure 1: the "lower" and "upper" decomposition should also contain a third "surface" contribution.

We will add this contribution in the right column in row five as "theta lower-boundary"

[Figure]

- Line 154: "30 days time slice"- 30 day time slice?

We will modify this sentence to: 30 day time slice

- Line 163: How is the separation defined? is it 600-200 hPa for the UPV, 600-875 hPa for the LPV, and 875hPa for the lower boundary? You should write this explicitly. Also, why do you take the 875 hPa as the lower boundary for the PPVI? Can't you use the surface temperature (or T2m temperature)? You later mention that you interpolate to 1000-50hPa, so why not use the lowest level? This might influence the PPVI composites.

a) The paragraph (lines 158-163) will be rewritten:

Furthermore, PV anomalies q' in upper- and lower-tropospheric layers are considered separately from each other (right column in Fig. 1): the upper layer between 550 and 150 hPa, the lower layer between 850 and 600 hPa and the lower boundary contribution at 875 hPa. Note that in the lower layer contribution, the PV anomalies and potential temperature at the boundary are separated, but the upper layer contribution contains both, PV anomalies and temperature at the upper boundary. The separation level between these anomalies (600 hPa) is consistent with the transition of the shape of the PV features from lower to higher levels (see Supplementary Fig. S2). See Fig. 1 for a general overview of the PPVI algorithm.

b) Due to local temperature variations at and near the surface, the lower boundary condition becomes noisier when using 1000 hPa instead of 875 hPa, which can compromise the convergence of the PV inversion. Using 875 hPa as lower boundary is also consistent with previous studies (e.g., Teubler F. and Riemer M., 2015).

- Line 166: "Following (Davids and Emanuel, 1991)..."- Following Davids and Emanuel (1991),...

We will modify this sentence to: Following Davis and Emanuel (1991),

- Figure 3: The switching of the colormap from panel (a) to panels (c) and (e) is really confusing! Please use the same colormap, and the same color to denote the same sign (e.g., blue to denote negative values, and red to denote positive values).

Fig. 3 will be modified so that all use the same color bar.

- Lines 270-275: The RV850 changes (Fig.5b) essentially show a more SW-NE elongated low level feature. Could this be a signature of enhanced Anticyclonic Wave Breaking (AWB), which is projected in the future in the region?

Our intense cyclones are generally characterized by cyclonic rather than anti-cyclonic wave breaking in the upper troposphere (Fig. 7c). But, consistent with your comment, this cyclonic wave breaking is projected to weaken in the future (Fig. 7d), which may also influence the lower-tropospheric vorticity pattern shown in Fig. 5b. Nevertheless, as this vorticity change has a relatively complex spatial pattern, we find it difficult to speculate about such a relationship to wave breaking.

- Figure 6: Why does PV increases everywhere in the troposphere? If these changes are due to enhanced LHR than I would expected positive PV changes below ~700-800 hPa, and negative PV changes above. This is not what you get.

Our hypothesis is that the amplified future LH in intense cyclones does not only enhance the low-level PV anomaly, but also leads to a larger vertical extent of this anomaly into the middle troposphere. This is consistent with previous results of Pfahl et. al. (2015, their Fig. 13) and Büeler and Pfahl (2019, their Fig. 5). Büeler and Pfahl

also show, based on a quantitative diagnostic relating LH and PV production, that such an upward shift can be expected due to enhanced LH.

In addition, we do not see any negative anomalies in these PV profiles because the outflow region of ascending air streams (WCBs) is located far away from the cyclone center (see our response to the following comments).

- Lines 341-349 (about the upper level PV decrease): your findings about the upper level PV reduction are confusing. First, the spatial distribution of the changes (Fig.7d) are confusing- they are not similar to other studies (e.g., Michaelis 2017 see their Fig. 8g,h), who find a PV decrease to the northeast of the UPV center (opposite from what you find!). The regions to the northeast (e.g., where the low level PV increases, Fig.7b) is where I would expect to find a PV decrease at upper levels, so I am confused. Could this reduction of PV that you find be just a signature of the upward lifting of the troposphere and tropopause? Can you plot vertical cross sections (without the spatial averaging) of PV and PV changes? Perhaps the peak of the PV maximum at upper levels has just shifted slightly upward?

When comparing our results to Michaelis et al. (2017), one should look at their Figure 7, which also shows intense (and not moderate) cyclones. Furthermore, their panels g and h show times prior to maximum cyclone intensity. The most appropriate comparison can thus be made between their Fig. 7i and our Fig. 7d. These two figures are actually very consistent, both showing PV decrease south and west of the cyclone center and an increase (although not significant in both analyses) to the northeast. We thus do not think that there is a discrepancy between our findings and previous studies.

Figure C1 below shows a PV cross section in present day climate and its future change, averaged over intense cyclones. While PV increases throughout the warm sector in the entire lower and middle troposphere, a PV decrease is visible around the tropopause level, which dominates the PV composite change at 250 hPa shown in Fig. 7d. However, this decrease is not simply due to an upward shift of the dynamical tropopause, as indicated by the green lines in Fig. C1.

We agree with the reviewer that one might expect a signature of reduced upper-level PV associated with the enhanced LH northeast of the cyclone center, in a region where the outflow of ascending WCB air streams can be expected. We hypothesize that we (and also Michaelis et al., 2017) do not see this signature in composites at this fully developed stage of the cyclones because the outflow has already spread out horizontally. This may shift the PV anomaly to other locations and also lead to smaller signals in the composite due to inconsistencies between the individual cyclones. In the second part of our study, we will investigate exactly this in more detail with the help of a Lagrangian methodology that is able to better capture such differences linked to the advection of PV anomalies.

In order to explain these complex issues more clearly in the paper, the discussion after line 344 (in the original manuscript) will be modified as follows:

"A similar reduction of upper-tropospheric PV has been found by Michaelis et al. (2017, their Fig. 7i). Parts of the PV decrease may be explained by changes in meridional PV advection. For instance, the slight increase in southerly upper-level wind velocities over the warm sector (see again Fig. 5f) may go along with enhanced advection of low-PV air masses from lower latitudes. In addition, an upward shift of the dynamical tropopause might contribute to such a PV decrease, although this tropopause shift is very small in the cyclones investigated here (see again Fig. 6) and thus not sufficient to explain the entire PV decrease. Finally, also enhanced LH at lower levels may impact the upper-level PV distribution through amplified upward motion and negative diabatic PV tendencies above the level of maximum heating. Michaelis et al. (2017) showed that this might lead to negative PV changes northeast of the cyclone center prior to (but not at) the time of maximum cyclone intensity. We will investigate the complex interplay between diabatic PV changes and PV advection (see also Brennan et al., 2008; Madonna et al., 2014) in the second part of this study with the help of a Lagrangian methodology."

[Figure]

Fig. C1. Average cross section (from west to east through the cyclone center, averaged over all intense cyclones) of the future PV change (shaded color) in the North Atlantic. The present-day mean PV distribution is shown as black contour lines. The tropopause is shown in the present-day (green solid line) and in the future climate (green dashed line). The composite shows the time of maximum intensity (time=0).

- Lines 354-359: Again, not clear to me. The diabatic heating from LHR usually tends to amplify the upper level ridge downstream of the positive PV anomaly, and this is not what you get.

This discussion will be removed/integrated in the text on upper-level PV changes above (see our previous response)

- Line 414: fairly way–> fairly well

We will modify this sentence to: fairly well

- Figure 10a: I find it confusing that the strongest meridional winds are to the west (i.e., southward), as I would expect, due to the induced winds from the UPV, to find the strongest v' to the east (i.e., poleward). Do you get the same results in the 850 hPa composites for the ERAI reanalysis data?

The upper-level contribution indeed leads to stronger southerly winds east of the cyclone center (Fig. 10b), but since both lower-level PV and the temperature anomaly at the lower boundary are associated with stronger northerlies to the west (Fig. 10 c,), this slightly dominates the complete pattern shown in Fig. 10a. Note, however, that the balanced wind (Fig. 10a) does not fully capture the complete wind pattern shown in Fig. 5c, which has its maximum southeast of the cyclone center, as discussed in lines 411-413.

A similar study with ERAI was done by Seiler (2019), who obtained similar results in terms of PV and THETA anomalies. Seiler (2019) does not show wind composites, but in his composites the maximum of relative vorticity at 850 hPa is also found slightly upstream of the cyclone center. Performing our PV inversion analysis again based on ERAI would be a substantial effort and, in our opinion, beyond the scope of this study.

- Figure 10d: Why is the wind velocity induced by the low level PV alone not more circular, as one would expect from the low level roughly circular PV anomaly?

The horizonal PV gradient is larger to the west than to the east of the cyclone center (Fig. 9a). This becomes clearer when Fig. 9a is plotted with a different color scale also showing lower PV values, we will thus adjust the figure accordingly.

[Figure]

Figure 9. Present-day composites for extreme cyclones of PV averaged over a) the lower troposphere (850-600 hPa), c) the upper troposphere (550-150 hPa) and potential temperature at 875 hPa (lower boundary) for winter in the North Atlantic region. Future changes of the lower tropospheric PV, upper tropospheric PV and potential temperature are shown in b, d, and f respectively. The present-day mean of each field is overlaid as black contour lines in b, d and f. The composites are shown at the time of maximum intensity (time=0).

- Figure 11: If you perform the PV inversion but not show the contributions from other layers, then this is not very meaningful (panels b and c are then just like taking a mean and perturbations). I think you should at least add the low level inversions to the SI.

As mentioned in the manuscript, the contributions from lower levels are very small. For completeness, we will add the contribution from lower-layer PV anomalies to the SI:

[Figure]

Figure S5. Wind composites at 250 hPa in present-day climate (a) and their future change (b). The wind composite is obtained from inverting the lower-layer PV anomalies.

---

## Author Comment (AC2)

**Answer to Review 2**

**General Comments**

The authors analyse the projected changes in cyclone intensity, PV anomalies and wind speed for North Atlantic cyclones in an 10-member ensemble of CESM-LENS climate simulations for the historical period (1990-2000) and late XXI Century following the RCP8.5 scenario (2090-2100). With this aim, a composite analysis is performed to evaluate the characteristics of the (most) extreme cyclones and how these are affected in a warmer climate. The main novelty of this study is the use of piecewise PV inversion to evaluate the relative contributions of PV changes at different levels to changes in low level winds, which in my opinion is a very promising approach (also to evaluate other cyclone features). The manuscript is well written and fits well into the scope of the journal. Moreover, it surely includes interesting and publishable material. Still, some aspects should be strengthened before the paper can be accepted for publication. I largely see these comments as "minor". Please find detailed comments below. If needed, I would be willing to review the paper again upon resubmission.

Thank you for providing a review for our manuscript and for your positive and helpful feedback. We have prepared this document to answer to your comments. The line number and figure references in the reviewer's comments refer to the original manuscript. The reviewer's comments are in black, and our responses are in blue

**Main Comments**

**a)** The main shortcoming in the present study is the limited discussion with the available literature, particularly with the "conclusions" section. This may have been postponed for the "part 2 manuscript", but as it is the manuscript has a bit of an unfinished feeling. For example, it would be helpful to clearly stated in how far the present manuscript provided new insights compared to recent review papers (notably Catto et al. 2019, also co-authored by S.P.)

Moreover, some more detailed discussion about the caveats of the selected approach would be helpful. Some statements are made within the results chapters (e.g. lines 200-202; 418-424), but these should be properly stated and discussed in the conclusions. This should include a) single model approach b) single tracking method c) selection of vertical levels d) PPVI decomposition

We will describe the limitations of this study in more detail and add more discussion in comparison with the literature. The respective part of the conclusion section (lines 472-500) will be supplemented as follows (original text in blue, new parts in red): At the end of the century, projected changes in cyclone frequencies are relatively small, with a general tendency towards slight decreases in many regions. Nevertheless, for the 10% most intense cyclones, an eastward displacement of the main oceanic storm track over the eastern North Atlantic is projected, associated with an increase in cyclone track density over northwestern Europe. These findings on cyclone frequency changes are generally consistent with previous studies using other climate models and cyclone tracking approaches (Pinto et al., 2009; Ulbrich et al., 2009; Zappa et al., 2013). Also, projected cyclone intensity changes, measured in terms of lower-tropospheric maximum relative vorticity or wind speed, are relatively small, again consistent with previous studies (Zappa et al., 2013).

In spite of such small overall intensity changes, our composite analysis indicates structural changes in the typical wind patterns associated with intense North Atlantic cyclones. In particular, an increase of wind velocities in the warm sector southeast of the cyclone center, potentially related to strengthening the low-level jet ahead of the cold front, and a southeastward broadening of the associated footprint of strong winds is projected. While some previous studies on future wind changes in cyclones have not detected such a robust change (Michaelis et al., 2017), consistent results regarding the broadening wind footprint have been obtained from idealized simulations (Sinclair et al., 2020) and a recent analysis of CMIP6 model projections (Priestley and Catto, 2021). Together with the eastward shift of storm tracks, this may lead to increased wind hazards in western Europe, which has also been seen in other model studies (Mölter et al., 2016).

In order to better understand the dynamical mechanisms behind these wind speed changes, a PV anomaly and inversion analysis have been conducted. PV inversion has been used previously to study future changes in cyclone propagation (Tamarin and Kaspi, 2017; Tamarin-Brodsky and Kaspi, 2017), but here it has been used for the first time for the investigation of future changes in the near-surface wind patterns associated with midlatitude cyclones. In agreement with many previous studies (Pfahl et al., 2015; Marciano et al., 2015; Michaelis et al., 2017; Zhang and Colle, 2018; Sinclair et al., 2020), we find an increase in lower-tropospheric PV near the cyclone center and fronts that is most likely due to increased latent heating in a warmer and thus more humid climate (Büeler and Pfahl, 2019). [...]

The analysis presented here has some limitations. It is based on a single climate model and thus does not take model uncertainty into account. Some confidence in the projection of the chosen CESM model is provided by the fact that the results on cyclone frequency changes and also the changes in near-surface wind patterns are consistent with other, multi-model studies (see again Ulbrich et al., 2009; Zappa et al., 2013; Priestley and Catto, 2021). On the other hand, by using several ensemble members, we have assessed the robustness of our findings with respect to natural climate variability (similar to, e.g., Yettella and Kay, 2017). Furthermore, our study uses a single cyclone tracking algorithm, which has been applied successfully before

in many other studies on midlatitude cyclones (e.g., Pfahl et al., 2015, Sprenger et al, 2017) and gives results that are in the range of other tracking schemes (Neu et al., 2013). Arguments for the robustness of our findings with respect to this choice of the tracking scheme are, again, that similar results have been obtained with other tracking algorithms, also using the same climate model (Day et al., 2016), and that the dependence on the tracking scheme is generally weaker for intense (compared to weak) cyclones (Neu et al., 2013, Ulbrich et al., 2013). Our results have been presented on specific vertical levels, but are generally robust with respect to small shifts of these levels (see for instance Figs. 8 and 9). Finally, as discussed in section 4.4, the PV inversion results can be affected by errors due to imperfect knowledge of boundary conditions, non-linearities and numerical inaccuracies. Especially the separation between low-level PV anomalies and lower boundary  $\theta$ -anomalies is affected, since the far impact of the low-level PV anomalies onto potential temperature below is not known. Nevertheless, we have shown that the associated residuum of the decomposition is relatively small and that the inversion method is able to reproduce the main features of the projected wind changes.

In summary, the PV analysis performed in this study provides insights into the role of altered upper-tropospheric dynamics and increased latent heat release in a warmer climate for future changes in near-surface wind fields around extratropical cyclones. The projected broadening of the wind footprint southeast of the cyclone center that can be explained by a combination of these processes may have important consequences for future changes in wind hazards. This study thus contributes to reducing the uncertainties associated with future changes in near-surface winds in cyclones (cf. Catto et al., 2019) through improved process understanding. In the second part of this study, Lagrangian air stream analyses will be used to complement and expand these dynamical insights.

**b)** The second main shortcoming is a limited quantification of uncertainty regarding the PPVI decomposition. While the uncertainty within the 10-member ensemble is shown in the previous sections and figures (e.g. line 343-344 regarding Fig. 7d), this is not the case for Figs 9-11. I wonder if this aspect could be enhanced (also in connection with lines 418-424).

The figures will be updated showing the agreement between ensemble members, as also shown below. The main characteristics (future response) described in the text are generally consistent for at least 80% of the ensemble members. A note on the relatively weak consistency of the low-level changes in the balanced flow (Fig. 10e) will be added to line 414:

Also, projected future changes in the balanced wind (Fig. 10e) reproduce changes in the full wind (Fig. 8b) fairly way, although they are less consistent between the different ensemble members.

Figure 9. Present-day composites for extreme cyclones of PV averaged over a) the lower troposphere (850-600 hPa), c) the upper troposphere (550-150 hPa) and potential temperature at 875 hPa (lower boundary) for winter in the North Atlantic region. Future changes of the lower tropospheric PV, upper tropospheric PV and potential temperature are shown in b, d, and f respectively. The present-day mean of each field is overlaid as black contour lines in b, d and f. The composites are shown at the time of maximum intensity (time=0). Green dots denote regions of ensemble agreement on the sign of change.

---

## Author Response (AR3)

**Reply to the Co-Editor**

We thank the co-editor Irina Rudeva for providing further comments on our revised manuscript. A point-by-point response to these comments can be found below.The co-editor's comments are in black and our response are in blue.

l.270-275,'poleward flow to the east enhances the poleward motion of the low-level cyclone, which can contribute to cyclone intensification when it crosses the upper-level jet axis Fig. 5e': If I got it right, cyclones were identified in the SLP field, meaning that the low-level center is already to the north of the jet axis.

Yes, you are right. We have modified this passage to make clearer what we mean:

The poleward flow to the east enhances the poleward motion of the low-level cyclone. Assuming that this poleward flow has also persisted in the period before maximum cyclone intensity, it may have contributed to cyclone intensification when the system crossed the upper-level jet axis (Riviere et al., 2013; Tamarin and Kaspi, 2017).

l.291, ' the wind increases are more robust than the decreases across the ensemble members': I don't think this is true for 5f. In 5d, the disagreement maybe a result of small shifts in the location of the negative anomaly between ensemble members. By the way, the color scale in 5d is either non-linear or suggests two digits after the decimal point.

a) We agree with your observation; in Fig. 5f, the wind increases are robust and wind decreases as well. We have modified the sentence  as below:

Note that, at low levels, the wind increases are more robust than the decreases across the ensemble members (see the green dots in Fig. 5d), which might be due to small shifts in the location of the negative anomaly between ensemble members.

b) The color bar in Fig 5d is not linear and it could be confusing.  Thus, we have decided to make it linear (every 0.2 m/s):

[Figure]

l.202: the months of October-March
Thank you, we have modified the sentence to:

… the months of October-March …

l.206:'averaged over a radial cap of a radius of 250 km ' -> averaged over a radius of 250 km

Thank you, we have modified the sentence to:

… averaged over a radius of 250 km …

363: using a Lagrangian approach

Thank you, we have modified the sentence to:

… in the second part of this study using a Lagrangian approach.

l.367: perhaps, 'complemented' is better; also, merge with previous paragraph.

Thanks, we have merged it with the previous paragraph and changed the sentence in line 367 to:

This qualitative discussion will be complemented by quantitative PPVI results below.

l.383: perhaps, 'upper-level PV future changes for extreme cyclones differ more substantially from those for intense cyclone'

Thanks, we have modified the sentence in line 383 to:

In contrast to the lower troposphere, upper-level PV changes for extreme cyclones differ more substantially from those for intense cyclone.